



# An experimental study of the reactivity of terpinolene and β-caryophyllene with the nitrate radical

Axel Fouqueau[1,2], Manuela Cirtog[1], Mathieu Cazaunau[1], Edouard Pangui[1], Jean-François Doussin[1], and Bénédicte Picquet-Varrault[1]

[1]Laboratoire Interuniversitaire des Systèmes Atmosphériques (LISA), UMR 7583, CNRS, Univ. Paris Est Creteil and Université de Paris, CNRS, LISA, F-94010 Créteil, France

[2]Laboratoire national de métrologie et d'essais (LNE), 75015 Paris, France

*Correspondance to:* Bénédicte Picquet-Varrault (benedicte.picquet-varrault@lisa.ipsl.fr)

**Abstract.** Biogenic volatile organic compounds (BVOCs) are subject to an intense emission by forests and crops into the atmosphere. They can rapidly react with the nitrate radical ($NO_3$) during nighttime to form number of functionalized products. Among them, organic nitrates (ON) have been shown to behave as reservoirs of reactive nitrogen and consequently influence the ozone budget and secondary organic aerosols (SOA) which are known to have a direct and indirect effect on the radiative balance, and thus on climate.

Nevertheless, BVOCs + $NO_3$ reactions remain poorly understood. Thus, the primary purpose of the follow-up study is to furnish new kinetic and mechanistic data for one monoterpenes ($C_{10}H_{16}$), terpinolene, and one sesquiterpene ($C_{15}H_{24}$), β-caryophyllene, using simulation chamber experiments. These two compounds have been chosen in order to fill the lack of experimental data. Rate constants have been measured using both relative and absolute methods. They have been measured to be $(5.5 \pm 3.8) \times 10^{-11}$ and $(1.7 \pm 1.4) \times 10^{-11}$ cm$^3$ molecule$^{-1}$ s$^{-1}$ for terpinolene and β-caryophyllene respectively. Mechanistic studies have also been conducted in order to identify and quantify the main reaction products. Total organic nitrates and SOA yields have been determined. Both terpenes appear to be major ON precursors both in gas and particle phase with formation yields of 69 % for terpinolene and 79 % for β-caryophyllene respectively. They also are major SOA precursor, with maximum SOA yields of around 60 % for both of the compounds. In order to support these observations, chemical analyses of the gas phase products were performed at the molecular scale using PTR-TOF-MS and FTIR. Detected products allowed proposing chemical mechanisms and providing explanations through peroxy and alkoxy reaction pathways.

## 1 Introduction

Human and biologic activities emit a large number of trace compounds into the atmosphere, including volatile organic compounds (VOCs). At the global scale, 90% of the VOCs are emitted by biogenic activities. Biogenic VOCs (BVOCs) include isoprene ($C_5H_8$), monoterpenes ($C_{10}H_{16}$), sesquiterpenes ($C_{15}H_{24}$) and oxygenated compounds (Guenther et al., 1995). Most of them are unsaturated VOCs and react rapidly with atmospheric oxidants leading to lifetimes below a minute for the most reactive ones. $NO_3$ radical has been shown to be an



efficient oxidant of these compounds during nighttime, but also during daytime under low sunlight conditions, e.g. below the forest canopy (Brown and Stutz, 2012).

These reactions lead to the formation of organic nitrates (ONs) which behave as reservoirs for reactive nitrogen by undergoing long-range transport in the free troposphere before decomposing and releasing NOx in remote regions (Ng et al., 2017). They therefore significantly influence the reactive nitrogenous species (NOy) and ozone budgets in these regions (Ito et al., 2007). Multifunctional organic nitrates are also expected to partition into condensed phases (aerosols, droplets) and this was confirmed by field observations which have shown that organic

nitrates range from 10% to 75% of total organic aerosol (OA) mass (Kiendler-Scharr et al., 2016a; Lee et al., 2016; Xu et al., 2015). ONs are therefore important components of OAs. A good understanding of the reactions of BVOCs and $NO_3$ is thus necessary to better assess the impact of these processes on air quality and radiative forcing. Nevertheless, for a number of BVOCs, this chemistry remains poorly studied.

In this study, we have investigated the reactivity of $NO_3$ radical with two BVOCs, terpinolene (a monoterpene)

and β-caryophyllene (a sesquiterpene) (see Fig. 1), using simulation chambers for determining both rate constants and mechanisms, with an experimental protocol similar to the one used in Fouqueau et al., 2020a:. Terpinolene represents 30% of Sassafras albidum monoterpene emissions and the global emission is estimated to 1.3 Tg year[-1] (Guenther et al., 2012). β-caryophyllene has been proved to be the most emitted sesquiterpene. It is also among the most emitted BVOCs by pine trees: it is the fifth most emitted compound by Pinus Taeda (3% of total

emissions, 47 identified species) and the second one by Pinus Virginiana (10% of total emissions, 34 identified species), with a global emission of 7.4 Tg year[-1]. Despite these two compounds have been detected in many tree emissions (Geron et al., 2000), their reactions with $NO_3$ radicals have been subject to a few studies only and little is known on this reactivity. Terpinolene has been subject to one absolute rate determination (Martinez et al., 1999) and two relative studies (Corchnoy and Atkinson, 1990; Stewart et al., 2013). The relative value measured by

Corchnoy and Atkinson, 1990 is almost 50 % higher than the other determinations. For β-caryophyllene, only one relative rate determination was conducted. No mechanistic study has ever been published for terpinolene to our knowledge, whereas two studies were published for β-caryophyllene: SOA yield was measured by Jaoui et al., 2013 and the chemical composition of the aerosol phase was analyzed. This study shows that β-caryophyllene + $NO_3$ is a major source of SOA, with a production yields estimated to 150%. Products in particle phase were

measured by collecting SOA on filters and by performing derivatisations followed by GC-MS analyses. Mass spectra observed for $NO_3$ oxidation were shown to be very different from those measured for other oxidants but no clear identification of the products was proposed. In addition, this study suggests that these products contain less nitrogen species than SOA from other terpenes (e.g. isoprene–$NO_3$ system). Fry et al., 2014 have also studied the SOA production from β-caryophyllene + $NO_3$. They have provided SOA yield plots and the organic nitrate

fraction in total aerosol mass. Nevertheless, the consumption of BVOC was very fast in this study and this could lead to an overestimation of SOA yields. For this reason also, some parameters, like ON yields, were not measured. New studies, both kinetic and mechanistic are necessary to have a better vision on the impact of these two compounds on air quality and radiative forcing.



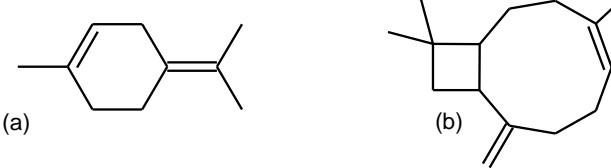

<p style="text-align:center">(a)          (b)</p>

**Figure 1: Molecular representation of terpinolene (a) and β-caryophyllene (b)**

**2 Experimental section**

The two different simulation chambers were used to study the reactions of terpinolene and β-caryophyllene with
NO$_3$ radicals: the CSA chamber and the CESAM chamber. Absolute and relative rate determinations were
conducted for both compounds. To tackle the determination of these very fast reactions, a highly sensitive
technique was requested for the monitoring of nitrate radicals. Absolute determinations were hence conducted
using an *in-situ* incoherent broadband cavity-enhanced absorption spectroscopy (IBBCEAS), which was recently
coupled with the CSA chamber (Fouqueau et al., 2020b). For both compounds, mechanistic studies have also been
conducted in CESAM chamber: total organic nitrate and SOA yields were determined and several individual gas-
phase products have been identified. Mechanisms have been proposed for the two compounds, using this
information.

**2.1 Chamber facilities and analytical devices**

Kinetic experiments were performed in the CSA chamber. It is a 6 meters long - 977 L - Pyrex® reactor (Doussin
et al., 1997) equipped with a homogenization system allowing a mixing time below one minute (Fouqueau et al.,
2020b). This chamber has been designed for the investigation of gas phase chemistry and is thus equipped with
instruments dedicated to gas phase monitoring. For measuring organic and inorganic species in the chamber, an
FTIR spectrometer (Bruker Vertex 80) is coupled to an *in situ* multiple reflection optical system. Spectra were
recorded with a resolution of 0.5 cm$^{-1}$, an optical path length of 204 m and a spectral range of 700-4000 cm$^{-1}$.

During absolute kinetic experiments, an *in situ* IBBCEAS technique was used to monitor NO$_3$ radicals at the ppt
level from its absorption at 662 nm. This technique is described in details in (Fouqueau et al., 2020b). It allows for
the monitoring of NO$_3$ radicals at very low concentrations (parts per trillion level) and exhibits a very good time
resolution (10 s). Simultaneously, it provides NO$_2$ concentration at the ppb level. Before each experiment, the
wavelength dependent mirror reflectivity, R(λ), has to be very precisely and accurately determined. For this
purpose, a known amount of NO$_2$ of several hundreds of ppb was introduced into the chamber. To quantify both
NO$_3$ and NO$_2$, cross sections were taken from Orphal et al., 2003 and Vandaele et al., 1997 respectively. At NO$_3$
maximum absorption (662.1 nm), the cross section is $(2.13 \pm 0.06) \times 10^{-17}$ cm$^2$ molecule$^{-1}$. Thanks to the very high
reflectivity of the mirrors (99.974 ± 0.002%), the maximum optical path length was calculated to be 2.5 km. This
configuration leads to a NO$_3$ detection limit of 6 ppt for 10 s of integration time. The relative uncertainty on NO$_3$
concentration was estimated to be 9%, with a minimum absolute value of 3 ppt (Fouqueau et al., 2020b).

To study the mechanisms and the SOA formation, experiments were carried out in CESAM chamber
(Experimental Chamber of Multiphase Atmospheric Simulation, (Wang et al., 2011) which has been specifically



designed to investigate multiphase processes. Briefly, it is a 4177 liters stainless-steel evacuable reactor equipped with a fan that allows for an efficient mixing within approximatively one minute (Wang et al., 2011). Aerosol

lifetimes in CESAM chamber are up to 3 days (depending on particle size – see supplementary material in Lamkaddam et al., 2017), which makes it well suited for SOA studies. The chamber is equipped with dedicated analytical instruments for gas and aerosol phases. To monitor the gas phase composition, an *in situ* long path FTIR spectrometer (Bruker Tensor 37) is coupled to the chamber. It allows measuring spectra in the 700-4000 $cm^{-1}$ spectral range with a resolution of 0.5 $cm^{-1}$ and an optical path of 174.5 m. A PTR-ToF-MS operating in both $NO^+$

and $H_3O^+$ ionization modes was also connected to the chamber. For the aerosol phase, a Scanning Mobility Particle Sizer (SMPS) composed of a TSI Classifier model 3080 and Differential Mobility Analyzer (DMA) model 3081 coupled to a Condensation Particle Counter (CPC) TSI model 3772 allows the measurement of the particle size distribution between 20 and 880 nm. Size distribution being measured in number, a particle density of 1.4 g $cm^{-3}$ was used to convert them into mass distribution (Fry et al., 2014; Draper et al., 2015; Boyd et al., 2015).

Integrated band intensities used in this study to quantify species of interest using FTIR are (in cm molecule$^{-1}$, logarithm base 10): $IBI_{terpinolene}$ (750-850 $cm^{-1}$) = $(4.22 \pm 0.4) \times 10^{-19}$, $IBI_{\beta\text{-caryophyllene}}$ (840-920 $cm^{-1}$) = $(1.60 \pm 0.2) \times 10^{-18}$, $IBI_{NO2}$ (1530-1680 $cm^{-1}$) = $(5.6 \pm 0.2) \times 10^{-17}$, $IBI_{HNO3}$ (840-930 $cm^{-1}$) = $(2.1 \pm 0.2) \times 10^{-17}$, $IBI_{N2O5}$ (1205-1275 $cm^{-1}$) = $(1.7 \pm 0.1) \times 10^{-17}$. This technique was also used to measure the total organic nitrate concentration, considering that all organic nitrates absorb at 850 $cm^{-1}$ and that the intensity of this band is weakly affected by the

chemical structure of the ON. In this study, the $IBI_{ON}$ (900-820 $cm^{-1}$) = $(9.5 \pm 2.9) \times 10^{-18}$ cm molecule$^{-1}$ was used (Fouqueau et al, 2020a).

In addition, a high resolution Proton Transfer Reactor – Time-of-Flight – Mass Spectrometer (PTR-ToF-MS) (Kore Series 2e, mass resolution of 4000) was used, both in $H_3O^+$ and $NO^+$ ionization mode. With $H_3O^+$ ionization mode, i.e. the standard operational conditions, organic nitrates have been shown to be subject to important

fragmentation (Müller et al., 2012; Aoki et al., 2007). To limit this, the electric field in the reactor has been reduced following the protocol proposed by Duncianu et al., 2017. The instrument was also operated in $NO^+$ ionization mode by using dry air instead of water as ionization gas, and by applying a reduced electric field in the reactor. In this mode, ONs are mainly ionized by charge transfer and by the formation of an adduct with $NO^+$, hence being detected at their own mass M and at M+30. Hydroxynitrates are a particular case, as they are detected at M-1

suggesting an ionization process involving a hydrogen loss.

To measure total ON yield in the aerosol phase, filter sampling was proceeded during experiments. Following a protocol described by Rindelaub et al., 2015, filters were extracted in 5 mL of $CCl_4$ and then analyzed with FTIR. Using two standards of organic nitrates (nitrooxypropanol and *tert*-butyl nitrate), they were quantified with IBIs of 510 and 580 L mol$^{-1}$ cm$^{-2}$ respectively between 1264 and 1310 $cm^{-1}$. Integrated absorption cross section of

organic nitrates in liquid phase was found to be $IBI_{ONs}$ (1264-1310 $cm^{-1}$) = $557 \pm 110$ L mol$^{-1}$ cm$^{-2}$, the difference between the IBIs measured for the two compounds being smaller than the uncertainty.

In order to compensate the decrease of pressure due to the instrument sampling, synthetic air was continuously introduced into the chamber during the course of the experiment. The mixture is then subject to a progressive dilution: for a typical flow rate of 1.7 L.min$^{-1}$, the dilution rate reaches max. 20% after 3 hours of experiment.

Therefore, all data presented here were corrected from dilution. The precise dilution rate is recorded on a one-





minute basis and these data are used for the dilution correction. SOA measurements were also corrected from particles physical wall loss which was parametrized as a function of the diameter of the particles and interpolated using the Lai and Nazaroff, 2000 model (friction velocity u*=3.7 cm s$^{-1}$, from Lamkaddam et al., 2017. In CESAM chamber, the wall loss appears to be very small (in comparison to Teflon chambers) thanks to stainless steel walls

that limit losses due to electrostatic effects.

### 2.1 Chemicals

Terpinolene and β-caryophyllene were purchased from Sigma-Aldrich at 95% and 98% of purity respectively. Synthetic air to fill the chambers was generated using 80 % of $N_2$ from liquid nitrogen evaporation (purity > 99.995 %, $H_2O$ < 5 ppm, Messer) and 20 % of $O_2$ (quality N5.0, purity > 99.995 %, $H_2O$ < 5ppm, Air Liquide).

$NO_3$ radicals were generated *in situ* from using the thermal dissociation of $N_2O_5$ which was first synthesized in a vacuum line from a protocol adapted from (Atkinson et al., 1984a; Schott and Davidson, 1958) and detailed in (Picquet-Varrault et al., 2009). The synthesis proceeds in two steps: first $NO_3$ is formed by the reaction between $O_3$ and $NO_2$ (Eq. 1) and then reacts with $NO_2$ to form $N_2O_5$ (Eq. 2). After a purification stage by pumping the bulb containing $N_2O_5$ for few minutes, it is introduced into the chamber and decomposes to form $NO_3$ radicals

(Reaction. 3).

$$O_3 + NO_2 \rightarrow NO_3 + O_2 \tag{1}$$
$$NO_3 + NO_2 + M \rightarrow N_2O_5 + M \tag{2}$$
$$N_2O_5 + M \leftrightharpoons NO_3 + NO_2 + M \tag{3}$$

### 2.3 Kinetic study

Kinetic experiments were conducted in the CSA chamber at room temperature and atmospheric pressure, in a mixture of $N_2/O_2$ (80/20). Both relative and absolute rate methods were used for an accurate determination of the rate constants. For absolute rate determination, PTR-ToF-MS and IBBCEAS were used to monitor BVOC and $NO_3$ concentrations respectively. Experiments were conducted by first introducing several hundred ppb of $NO_2$

into the chamber in order to determine the reflectivity of the IBBCEAS mirrors. Then, the BVOC was injected and left in the dark for approximatively one hour. It allows checking for eventual wall loss or reaction with $NO_2$. No significant loss was observed. In order to avoid large SOA formation that would strongly reduce the IBBCEAS signal due to light absorption/scattering and mirror soiling by particles, low BVOC mixing ratios have been used (between 15 and 90 ppb). Finally, $NO_3$ was generated *in situ* (see section 2.1) by stepwise injections of $N_2O_5$ and

measurements were performed with a time resolution of 10 seconds in order to allow monitoring fast decay of reactants. Several stepwise injections of $N_2O_5$ were made until the complete BVOC consumption.

Considering the following reaction,

$$BVOC + NO_3 \rightarrow Products \tag{4}$$

the second-order kinetic equation is obtained:

$$-\frac{d[BVOC]}{dt} = k_{BVOC}[BVOC][NO_3] \tag{5}$$

For small time intervals, it can be approximated as:





$$-\Delta[BVOC] = k_{BVOC}[BVOC][NO_3]\Delta t \qquad (6)$$

where $-\Delta[BVOC]$ is the decay of BVOC during $\Delta t$ time interval and $[BVOC]$ and $[NO_3]$ are averaged concentrations during this interval. $k_{BVOC}$ is obtained by plotting $-\Delta[BVOC]$ vs $[BVOC] \times [NO_3] \times \Delta t$. It should

be mentioned that the determination of the rate constant is thus not affected by losses of $NO_3$ due to reaction with other species (products, $RO_2$ radicals, etc.) as the rate constant is not deduced from $NO_3$ consumption rate but from the BVOC one. The uncertainty on $k_{BVOC}$ was taken as twice the standard deviation on the slope.

For relative rate determination, PTR-ToF-MS and FTIR techniques were used to monitor the BVOC decay relatively to a reference compound. As for absolute rate experiments, the organic reactants were left in the dark

for one hour prior to the $N_2O_5$ injection. By assuming that consumption by $NO_3$ is the only fate of the studied BVOC and the reference compound, and that these compounds are not a product of both of the reactions, the following equation can be shown (Atkinson, 1986):

$$ln\left(\frac{[BVOC]_{t_0}}{[BVOC]_t}\right) = \frac{k_{BVOC}}{k_{Ref.}} ln\left(\frac{[Ref.]_{t_0}}{[Ref.]_t}\right) \qquad (7)$$

where $[BVOC]_{t_0}$ and $[Ref.]_{t_0}$ are BVOC and Ref. concentrations at time $t_0$ (which correspond to the moment before the beginning of the oxidation), $[BVOC]_t$ and $[Ref.]_t$ are the concentrations at $t$ time and $k_{BVOC}$ and $k_{Ref.}$ are the rate constants with $NO_3$ respectively.

In this work, 2,3-dimethyl-2-butene was used as reference compound because of its well-known rate constant with $NO_3$ radicals. Due to the lack of recommendation by IUPAC for this reaction, its rate constant was calculated as

the mean value of the determinations available in the literature (Berndt et al., 1998; Benter et al., 1992; Lancar et al., 1991; Rahman et al., 1988; Atkinson et al., 1988, 1984a, b). The obtained value is: $k_{2,3\text{-dimethyl-2-butene}} = (5.5 \pm 1.7) \times 10^{-11}$ $cm^3$ $molecule^{-1}$ $s^{-1}$. The uncertainty on the rate constant was calculated as twice the standard deviation of all the values. Finally, the uncertainty on $k_{BVOC}$ was calculated by considering the relative uncertainty corresponding to the statistical error on the linear regression ($2\sigma$) and the error on the reference rate constant.


### 2.4 Mechanistic study

Mechanistic study was conducted in CESAM chamber at room temperature and atmospheric pressure, in a mixture of $N_2/O_2$ (80/20). Experiments were typically conducted by first introducing the BVOC into the chamber and leave it in the dark approximatively one hour to estimate possible wall losses. No significant wall loss was observed for

both studied BVOCs ($k_d < 10^{-7}$ $s^{-1}$). Then $N_2O_5$ was introduced by slow continuous injections as this method has been observed to be more efficient than stepwise injections to slow down the oxidation, and thus to better control the SOA formation. PTR-ToF-MS and FTIR spectrometer were used to monitor both BVOC and gas phase products. In some experiments, two PTR-ToF-MS were used in order to detect gas-phase products in both $NO^+$ and $H_3O^+$ ionization modes simultaneously. If using two instruments was not possible, experiments were

duplicated. SMPS was used to monitor the SOA production. Because of the lack of standards, quantification of gas-phase products measured by PTR-ToF-MS was not possible. In order to measure SOA yields under low aerosol content, no seed particles were introduced into the chamber. Filter sampling was performed for experiments for which the concentration of precursor was up to 150 ppb. It started when the precursor has completely reacted and lasted for 3 to 6 hours. To avoid the condensation of gas phase products on the filter, a charcoal denuder was used.





Products formation yields were calculated by plotting the molecular concentration of product against the reacted BVOC molecular concentration and by calculating the slope of the straight line. To calculate the total organic nitrate yields in SOA phase, final organic nitrates concentration measured on the filters was divided by the total reacted BVOC concentration. Uncertainty on the yield was calculated as the sum of the relative uncertainties on organic nitrates and BVOC concentrations.

SOA yield is defined as the ratio between the produced SOA mass concentration, $M_0$, and the reacted BVOC mass concentration, $\Delta$BVOC. It was calculated for each data point and after the total consumption of the BVOC for all experiments, providing time-dependent and overall SOA yields. They were plotted against the organic aerosol mass and a fit was applied using a two-product model described by Odum et al., 1996:

$$Y = M_0 \left[ \frac{\alpha_1 K_{p,1}}{1 + K_{p,1} M_0} + \frac{\alpha_2 K_{p,2}}{1 + K_{p,2} M_0} \right]$$  (8)

Where $\alpha_1$, $\alpha_2$ and $K_{p,1}, K_{p,2}$ are stoichiometric factors and partitioning coefficients (in m$^3$ µg$^{-1}$) of the two hypothetical products respectively. It was expected that SOA equilibrium was reached at small time steps because of the slow injections of N$_2$O$_5$, thus time-dependent yields have been used. Hence, yields for small aerosol content have also be obtained.

In order to assess their contribution to SOA formation, vapor pressures $P^{vap}$ have been evaluated using SIMPOL-

1 method (Pankow and Asher, 2008) via the GECKO-A website (http://geckoa.lisa.u-pec.fr, last access Mars 05$^{th}$ 2021). In order to estimate the fraction of a product i in the condensed phase $\xi_{aer}^i$, Raoult's law has been used (Valorso et al., 2011):

$$\xi_{aer}^i = \frac{N_{i,aer}}{N_{i,aer} + N_{i,gas}} = \frac{1}{1 + \frac{\overline{M_{aer}} \gamma_i P_i^{vap}}{C_{aer} RT} \times 10^6}$$  (9)

where N$_{i,gas}$ and N$_{i,aer}$ are respectively the gas and particle phase concentrations (in molecule cm$^{-3}$) of the product

i, $\overline{M_{aer}}$ the SOA species mean molecular weight (g mol$^{-1}$), C$_{aer}$ is the total SOA mass concentration (µg m$^{-3}$), R the gas constant (atm m$^3$ K$^{-1}$ mol$^{-1}$), T the temperature (K), P$_i^{vap}$ the vapor pressure and γ$_i$ the product i activity coefficient (γ$_i$ = 1 was used in this study). Here, $\overline{M_{aer}}$ has been estimated to be the mean molecular weight of detected low volatility products.

The calculation of $\xi_{aer}^i$ depends strongly on the estimation of $P^{vap}$. It was shown by Pankow and Asher, 2008 that

SIMPOL-1 technique predicts it with an uncertainty between 50 % and 60 % for $P^{vap} < 10^{-6}$ atm. $\xi_{aer}^i$ can only be used as a guide, because it is associated with a high uncertainty. $\xi_{aer}^i$ has also be compared to the partitioning coefficients $K_p$ used in Eq. (8):

$$K_p = \frac{N_{i,aer}}{N_{i,gas}} \times \frac{1}{C_{aer}} = \frac{\xi_{aer}^i}{1 - \xi_{aer}^i} \times \frac{1}{C_{aer}}$$  (10)

**3 Kinetic results**

All experiments and their conditions are presented in Table 1. Both of the compounds were subject to absolute and relative rate determinations. For each method, between two and four experiments were conducted.





**Table 1: Experimental conditions of kinetic experiments. [BVOC]$_i$ and [Ref.]$_i$ are the initial mixing ratios of BVOC and of the reference compound. For [N$_2$O$_5$], the number of punctual injections is indicated in brackets. T is the mean temperature inside the simulation chamber during the experiment.**

| BVOC | Date (yyyy/mm/dd) | Method* | [BVOC]$_i$ (ppb) | Ref. | [Ref.]$_i$ (ppb) | [N$_2$O$_5$]$_i$ (ppb) | [NO$_2$]$_i$ (ppb) | T (K) |
|---|---|---|---|---|---|---|---|---|
| Terpi. | 2018/04/25 | RR | 190 | 2,3-dimethyl-2-butene | 210 | 200 (2 inj.); 300 | - | 295.15 |
| | 2018/04/26 | RR | 540 | 2,3-dimethyl-2-butene | 420 | 100; 200 (2 inj.); 300 | - | 294.55 |
| | | RR | 240 | 2,3-dimethyl-2-butene | 330 | 300 (3 inj.); 400 | - | 294.85 |
| | 2018/04/24 | AR | 15 | - | - | 20 (2 inj.) | 750 | 295.45 |
| | | AR | 31 | - | - | 40 (2 inj.) | 740 | 295.55 |
| | 2018/04/25 | AR | 45 | - | - | 40 (2 inj.) | 890 | 295.15 |
| β-car. | 2018/04/20 | RR | 550 | 2,3-dimethyl-2-butene | 360 | 200; 300 (2 inj.); 600 | - | 295.45 |
| | | RR | 690 | 2,3-dimethyl-2-butene | 650 | 200; 400 (2 inj.); 600 | - | 295.45 |
| | 2018/04/18 | AR | 45 | - | - | 20 (3 inj.) | 540 | 294.85 |
| | | AR | 86 | - | - | 20 (3 inj.) | 560 | 295.05 |
| | 2018/04/19 | AR | 67 | - | - | 20 (2 inj.) | 560 | 295.35 |
| | | AR | 36 | - | - | 20 (2 inj.) | 590 | 295.55 |

* RR: relative rate determination; AR: absolute rate determination

Kinetic results obtained by the relative rate method are plotted in Fig. 2. They present good linear tendencies and are in good agreement whatever the analytical technique used. For both individual data sets obtained by PTR-ToF-

MS and FTIR, linear regressions have been first performed separately. The results being in good agreement, a global linear regression was applied to all the data, leading to $k_{terpinolene} = (6.0 \pm 2.5) \times 10^{-11}$ cm$^3$ molecule$^{-1}$ s$^{-1}$ and $k_{\beta-caryophyllene} = (1.4 \pm 0.7) \times 10^{-11}$ cm$^3$ molecule$^{-1}$ s$^{-1}$.

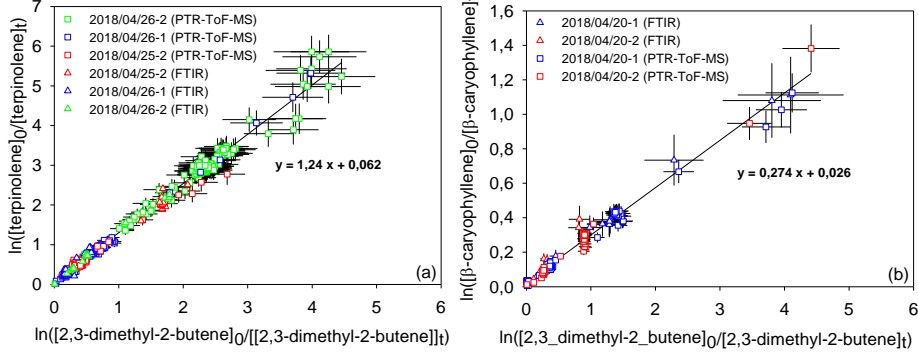

**Figure 2: Relative kinetic plots measured by FTIR (triangle marks) and PTR-ToF-MS (round marks) for**

**terpinolene (a) and β-caryophyllene (b).**





Absolute kinetic plots are shown in Fig. 3. Experimental points are rather scattered and this can be explained by the low integration time used for PTR-ToF-MS and IBB-CEAS measurements. As a consequence, rate constants are subject to relatively high uncertainties. Rate constants measured by the absolute rate method are: $(4.9 \pm 1.4) \times 10^{-11}$ cm$^3$ molecule$^{-1}$ s$^{-1}$ for terpinolene and $(2.0 \pm 0.6) \times 10^{-11}$ cm$^3$ molecule$^{-1}$ s$^{-1}$ for β-caryophyllene.

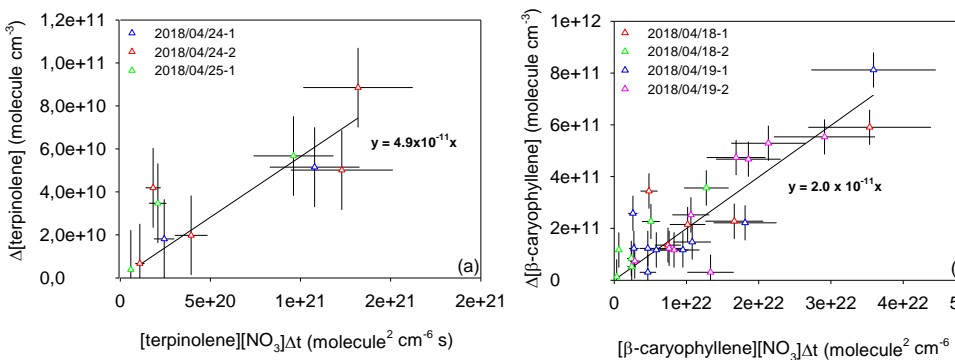


**Figure 3 : Absolute kinetic plots for terpinolene (a) and β-caryophyllene (b).**

The values measured by both the relative and the absolute methods are compared to those already published in the literature in Table 2. In order to compare our value with literature data, relative rate from Corchnoy and Atkinson, 1990 has been recalculated using the same value for the reference rate constant (see section 2.3). The value obtained by Stewart et al., 2013 with limonene as reference compound, was also recalculated using the latest IUPAC recommendation: $(1.2 \pm 0.4) \times 10^{-11}$ cm$^3$ molecule$^{-1}$ s$^{-1}$. Finally, the rate constant provided by (Shu and Atkinson, 1995) was recalculated using the value $(9.6 \pm 1.6) \times 10^{-11}$ cm$^3$ molecule$^{-1}$ s$^{-1}$ for 2-methyl-2-butene. This value has been obtained by averaging all determinations published in the literature. The total uncertainties presented in this table for relative rate determinations are the sum of the statistical errors provided by the authors and the errors on the reference rate constants.

**Table 2 : Rate constants for the NO$_3$-initiated oxidation of terpinolene and β-caryophyllene: results from this study and comparison with literature. Rate constants of α- and γ-terpinene found by Fouqueau et al., 2020a are also shown.**

| BVOC | k (cm$^3$ molecule$^{-1}$ s$^{-1}$) | (k$_{COVB}$/k$_{ref}$) | Study (method) |
|------|-------------------------------------|------------------------|----------------|
| | $(4.9 \pm 1.4) \times 10^{-11}$ | | **This study (AR[a])** |
| Terpinolene | $(6.0 \pm 2.5) \times 10^{-11}$ | $(1.2 \pm 0.1)$ | **This study (RR[b] : 2.3-dimethyl-2-butene)** |
| | $(8.5 \pm 3.4) \times 10^{-11}$ | $(1.7 \pm 0.1)$ | Corchnoy and Atkinson, 1990 (RR[b] : 2.3-dimethyl-2-butene) |





| | | | |
|---|---|---|---|
| | $(5.2 \pm 0.9) \times 10^{-11}$ | | Martinez et al., 1999 (AR[a]) |
| | $(6.2 \pm 3.0) \times 10^{-11}$ | $(5.1 \pm 0.4)$ | Stewart et al., 2013 (RR[b] : Limonene) |
| | $6.6 \times 10^{-11}$ | | Estimated with SAR (Kerdouci et al., 2014) |
| β-caryophyllene | $\mathbf{(2.0 \pm 0.6) \times 10^{-11}}$ | | **This study (AR[a])** |
| | $\mathbf{(1.4 \pm 0.7) \times 10^{-11}}$ | $\mathbf{(0.27 \pm 0.04)}$ | **This study (RR[b] : 2,3-dimethyl-2-butene)** |
| | $(2.0 \pm 0.7) \times 10^{-11}$ | $(2.1 \pm 0.4)$ | **Shu and Atkinson, 1995 (RR[b] : 2-methyl-2-butene)** |
| | $2.1 \times 10^{-11}$ | | Estimated with SAR (Kerdouci et al., 2014) |
| γ-terpinene | $(2.9 \pm 1.1) \times 10^{-11}$ | | Fouqueau et al., 2020a |
| α-terpinene | $(1.2 \pm 0.3) \times 10^{-10}$ | | Fouqueau et al., 2020a |

[a]: Absolute rate determination; [b]: Relative rate determination

For terpinolene, the absolute and relative determinations obtained in this work are in good agreement. They also appear to be in good agreement with the values provided by previous studies, within uncertainties. Nevertheless, when considering the values $k_{COVB}/k_{ref}$ obtained in this work and by Corchnoy and Atkinson, 1990 (both were using the same reference compound), it appears that the two relative rate determinations are not in agreement. The

value obtained by Corchnoy and Atkinson, 1990 is 40% higher than our study. No explanation has been found for this difference, but it can be seen that the value of Corchnoy and Atkinson, 1990 is higher than every other value in the literature. Our result thus confirms the lower values found by previous studies. For β-caryophyllene, the absolute and relative determinations obtained in this work are also in good agreement. These determinations have been compared to the only determination previously published by Shu and Atkinson, 1995. A good agreement can

be observed, whatever the method used. Our study provides the first absolute rate determination for β-caryophyllene. Our data were also compared to estimated rate constants using the structure-activity relationship (SAR) developed by Kerdouci et al., 2014. Experimental and estimated rate constants show a good agreement.

Terpinolene rate constant can be compared to the values found by Fouqueau et al., 2020a for α-terpinene and γ-terpinene and shown in Table 2. They indeed have very similar structures, only differing by the position of the

double bounds. α- and γ-terpinene have endocyclic double bonds (respectively, conjugated and not conjugated), whereas terpinolene has one endocyclic and one exocyclic double bonds. α-terpinene appears to be much more reactive than γ-terpinene, due to the conjugation of double bonds: after $NO_3$ addition onto one of the double bonds, the alkyl radical formed is indeed stabilized by the delocalization of the single electron. Here, terpinolene is almost twice more reactive than γ-terpinene, and this can be explained by the fact that the exocyclic double bond is

composed of two quaternary carbons which are more stable than the tertiary carbons of γ-terpinene. In addition, terpinolene which has non-conjugated C=C bonds is less reactive than α-terpinene.

### 4 Mechanistic results

Seven mechanistic experiments were conducted in CESAM chamber for terpinolene and nine for β-caryophyllene. During experiments, the formation of gas-phase products and SOA was monitored. Table 3 presents experimental

conditions together with organic nitrates and SOA yields that were measured. As an example, reactants and products time-profiles (corrected from dilution) are presented in Fig. 4 for the experiment 2017/12/18 on terpinolene. In the first minutes following $N_2O_5$ injection (squared by the red area), a competition occurs between the reactivity of $NO_3$ on BVOC and its wall loss through $N_2O_5$ hydrolysis. In the beginning of the experiment,


mainly nitric acid is thus formed by $N_2O_5$ hydrolysis on lines and chamber walls. Then, the BVOC starts to be
oxidized with the weakening of the hydrolysis reaction. Because small quantities of $N_2O_5$ were introduced
continuously in order to ensure a progressive oxidation of the BVOC, $N_2O_5$ concentration remains below the
detection limit as long as the BVOC is not totally consumed (around 25 minutes here). The formation of large
amounts of organic nitrates and SOA are observed: for an initial terpinolene mixing ratio of 180 ppb, up to 70 ppb
of total organic nitrates and 400 µg/m$^3$ of aerosol are formed. Figure 4 shows also the aerosol size distribution. It
can be seen that particles have mean diameters around 300-400 nm. PTR-ToF-MS signals (m/z) time profiles are
presented in Fig. S1 and are discussed later with their identification.

For β-caryophyllene, only two experiments could be used to determine the SOA yields. Indeed, except for
experiments conducted in December 2017, very large amounts of SOA were formed (between 500 µg/m$^3$ and 1
mg/m$^3$) and the upper part of the size distribution fell out of SMPS range affecting the relevance of the mass
evaluation from SMPS measurement.



**Table 3: Experimental conditions, ONs and SOA yields for mechanistic experiments conducted in CESAM chamber. The use of an instrument is shown by a cross, the non-use by a dash.**

| BVOC | Date (yyyy/mm/dd) | $[BVOC]_i$ (ppb)* | $N_2O_5$ injection (concentration and/or duration) | PTR-ToF-MS ($NO^+$) | PTR-ToF-MS ($H_3O^+$) | Filter sampling & analysis | $Y_{acetone, molar}$ | $Y_{ONg, molar}$ | $Y_{ONp, molar}$ | $Y_{ON(g+p), molar}$ | $Y_{ONp, mass}$ | $Y_{SOA mass}$ | $Y_{ONp, mass} / Y_{SOA,mass}$ |
|---|---|---|---|---|---|---|---|---|---|---|---|---|---|
| Terpi-nolene | 2017/04/03 | 300 | Continuous (30 min) | x | - | x | 0.31 ± 0.03 | 0.54 ± 0.07 | 0.23 ± 0.08 | 0.77 ± 0.37 | 0.36 ± 0.14 | 0.64 ± 0.17 | 0.56 ± 0.37 |
| | 2017/04/21 | 350 | Continuous (35 min) | x | - | x | 0.25 ± 0.03 | 0.52 ± 0.04 | 0.17 ± 0.07 | 0.69 ± 0.34 | 0.31 ± 0.12 | 0.47 ± 0.17 | 0.66 ± 0.49 |
| | 2017/04/24 | 360 | Continuous (55 min) | - | x | x | 0.18 ± 0.02 | 0.41 ± 0.04 | 0.19 ± 0.07 | 0.60 ± 0.28 | 0.27 ± 0.10 | 0.63 ± 0.18 | 0.42 ± 0.28 |
| | 2017/12/12 | 48 | Continuous (17 min) | x | x | - | 0.21 ± 0.02 | 0.21 ± 0.01 | - | - | - | 0.25 ± 0.09 | - |
| | 2017/12/12 | 120 | Continuous (48 min) | x | x | - | 0.18 ± 0.02 | 0.25 ± 0.04 | - | - | - | 0.29 ± 0.09 | - |
| | 2017/12/13 | 120 | Continuous (23 min) | x | x | - | 0.11 ± 0.03 | 0.25 ± 0.02 | - | - | - | 0.33 ± 0.09 | - |
| | 2017/12/18 | 180 | Continuous (24 min) | x | x | x | 0.11 ± 0.03 | 0.30 ± 0.01 | 0.07 ± 0.03 | 0.37 ± 0.17 | 0.11 ± 0.04 | 0.41 ± 0.11 | 0.27 ± 0.17 |
| β-caryo-phyllene | 2016/12/14 | 490 | Continuous (11 min) | x | x | - | - | 0.68 ± 0.03 | - | - | - | exceed. range | - |
| | 2016/12/14 | 390 | Continuous (56 min) | x | x | - | - | 0.50 ± 0.01 | - | - | - | exceed. range | - |
| | 2016/12/15 | 530 | Continuous (60 min) | x | - | - | - | 0.48 ± 0.01 | - | - | - | exceed. range | - |
| | 2016/12/15 | 450 | Continuous (45 min) | x | - | - | - | 0.49 ± 0.01 | - | - | - | exceed. range | - |
| | 2016/12/16 | 410 | Continuous (12 min) | x | - | - | - | 0.57 ± 0.02 | - | - | - | exceed. range | - |
| | 2017/04/04 | 840 | Continuous (48 min) | x | - | x | - | 0.57 ± 0.03 | 0.21 ± 0.09 | 0.78 ± 0.37 | 0.31 ± 0.12 | exceed. range | - |
| | 2017/04/28 | 430 | Continuous (37 min) | - | X | x | - | 0.53 ± 0.01 | 0.24 ± 0.09 | 0.77 ± 0.30 | 0.35 ± 0.13 | exceed. range | - |
| | 2017/12/15 | 80 | Continuous (20 min) | x | X | x | - | 0.52 ± 0.09 | - | - | - | 0.51 ± 0.14 | - |
| | 2017/12/22 | 230 | Continuous (49 min) | x | X | x | - | 0.41 ± 0.01 | 0.25 ± 0.09 | 0.66 ± 0.25 | 0.34 ± 0.13 | 0.43 ± 0.11 | 0.79 ± 0.50 |

*for all experiments, the BVOC was totally consumed.


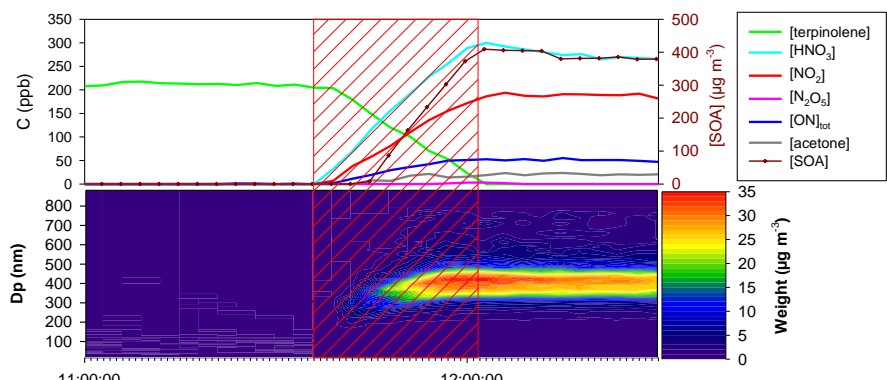

**Figure 4: Dilution-corrected time-dependent concentration of gaseous species, aerosol mass, SOA size distribution during a typical experiment of $NO_3$-initiated oxidation of terpinolene (2017/12/18). Red dashed area corresponds to $N_2O_5$ injection period. Top figure: terpinolene, $N_2O_5$, $NO_2$, $HNO_3$, acetone and total ONs from FTIR and SOA mass concentration from SMPS; bottom figure: SOA size distribution in mass concentration from SMPS.**

### 4.1 SOA Yields

Figure 5 shows time-dependent and overall SOA yields ($Y_{SOA}$) as a function of the aerosol mass ($M_0$) for both terpinolene and β-caryophyllene. As explained before (see Section 2.4), a two products model defined by Odum et al., 1996, has been applied for the two compounds. Final yields obtained for terpinolene can reach 60 % whereas they are below 90% for β-caryophyllene. These results demonstrate both of the compounds are very efficient SOA precursors.

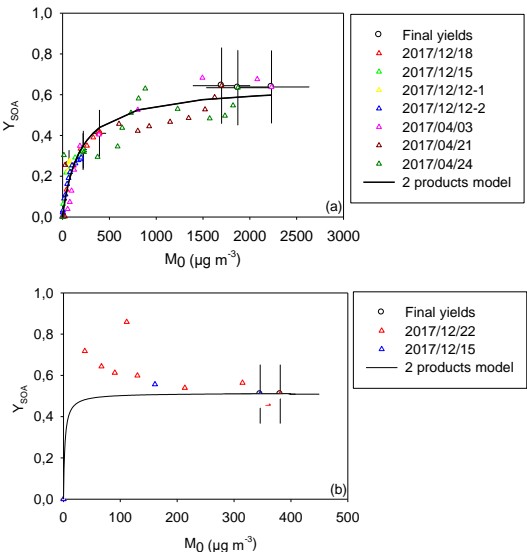

**360** **Figure 5: SOA yield as a function of the organic aerosol mass concentration measured for terpinolene (a) and for β-caryophyllene (b). Final yields (circle marks) are shown with uncertainties. Data were fitted with a two-product model (black curve).**

For terpinolene, our study provides the first determination of SOA yields. Fig. 5 shows that fitted plots are well constrained for small aerosol contents (below 50 µg m$^{-3}$) thanks to the high number of experimental points in this

**365** area. This is a consequence of the slow injection of $N_2O_5$ which allows a progressive BVOC oxidation. Fitted parameters have been found to be $\alpha_1 = 0.6$ ; $K_{p,1} = 6.7 \times 10^{-3}$ m$^3$ µg$^{-1}$ et $\alpha_2 = 3 \times 10^{-3}$ ; $K_{p,2} = 3.5 \times 10^{-1}$ m$^3$ µg$^{-1}$. The very low stoichiometric factor $\alpha_2$ indicates that the second class of products is negligible, so the particle phase products can be simulated with only one family of same volatility. One can estimate the uncertainties of fitting parameters by looking at the fit sensitivity. It appears to be very sensitive to $\alpha$ (with an associated error

**370** estimated to 5 %) and less to $K_p$ (with an error estimated to 50 %). For an aerosol mass concentration typical of a biogenic SOA affected environment of 10 µg m$^{-3}$ (Slade et al. 2017), SOA yield of 5 % has been measured for terpinolene. For higher aerosol mass loading, which can be observed in polluted atmospheres (between 500 and 1000 µg m$^{-3}$), yield reaches 50-60 %.

For β-caryophyllene, a high dispersion is observed between the data from the two experiments, for low aerosol

**375** mass loadings that correspond to the first stages of the oxidation. For the experiment 2017/12/22, an "unusual" profile is observed in the sense that the SOA yield decreases with the increasing $M_0$. This suggests that, despite a slow injection of $N_2O_5$, the oxidation of the β-caryophyllene was too fast, in comparison to the mixing time, leading to a locally high concentration of semi-volatile species and therefore, to an overestimation of the SOA yield. After few minutes, the SOA yield decreases and is then in good agreement with those measured for the experiment

**380** 2017/12/15, suggesting that semi-volatile species are better mixed in the reactor, and therefore SOA yields, are more accurate. The Odum fitting parameters obtained from these two experiments are: $\alpha_1 = 0.5$ ; $K_{p,1} = 4.1 \times$



$10^{-1}$ m³ µg⁻¹ and $\alpha_2 = 3.8 \times 10^{-3}$ ; $K_{p,2} = 5 \times 10^{-1}$ m³ µg⁻¹. As for terpinolene, the high value of $\alpha_1$ and low $\alpha_2$ one indicate that one class of products, having a high partitioning coefficient ($K_{p.1}$) contributes mainly to the SOA formation. These results also show that β-caryophyllene is a very efficient SOA precursor with a yield close to 40 % at 10 µg m⁻³ which can reach almost 60 % for higher aerosol mass loading. Nevertheless, due to the experimental problems mentioned above, this model is not well constraint for low aerosol mass loading (< 100 µg m⁻³) and these results have to be taken with caution. Two studies have been previously conducted on the SOA production from β-caryophyllene. First, Jaoui et al., 2013 measured SOA yields in a simulation chamber. In this study, final aerosol yield has been provided without indication on the aerosol mass loading, thus preventing from fitting data by the Odum model. Yields were shown to range between 91 and 146 %. Fry et al., 2014 study has provided SOA yields curves and ON yields in particle phase. This study has been conducted with high and low BVOC concentrations (respectively 3 and 109 ppb). Since experiments were carried out by introducing the oxidant into the chamber prior to the BVOC, this last one began to react immediately, preventing measurement of its initial concentration. The consumption of the BVOC had therefore to be estimated. In a similar way to our study, the authors have observed differences between high and low concentration mass yield curves suggesting that the experiments differ in more than simply the total aerosol mass loading. They measured higher yields for high concentration experiments than for low concentration experiments (for the same aerosol mass loading). The authors recommend using preferentially data obtained for low concentration experiments considering that due to the slower reaction, the ΔVOC is better constrained for longer periods and the mixing time scale is faster relative to reactions, resulting in more precise yield curves. However, even for these low concentration experiments, the yields obtained (around 80 %) are much higher than those measured in our study. Such disagreement could be explained by the fact that ΔVOC is not precisely measured in Fry et al., 2014 study. Another possible explanation provided by the authors that could explain the difference between high and low concentration experiments but also the disagreement between their results and our study, may lie in the differences in $RO_2$ radical fate. $RO_2$ radicals can indeed react following several pathways, in particular with $NO_3$ or with other $RO_2$ radicals and products resulting from these two reactions differ. For example, $RO_2+RO_2$ reactions can produce hydroxy-nitrates which have low volatility and can thus participate to SOA formation (see discussion in section 4.3). In conclusion, this discussion illustrates well how SOA yields may be affected by a number of parameters and how comparisons are difficult to interpret.

## 4.2 Organic nitrates yields

The total ON yields have been measured in the gas phase ($Y_{ONg}$). Their concentrations have been plotted against the consumption of BVOC for both of the studied compounds in Fig. 6. The plots show a good linearity and the slope at the origin is different from zero. This indicates that i) organic nitrates are primary products and ii) if they themselves react with $NO_3$ by addition to the other C=C bond, they produce secondary organic nitrates, so that the total ON yield is constant during the course of the experiments. Previous studies performed in CESAM chamber have reported that ONs may be subject to wall losses, through absorption on the stainless steel walls (Suarez-Bertoa et al., 2012; Picquet-Varrault et al., 2020). Loss rates have been found to be between 0.5 and $2 \times 10^{-5}$ s⁻¹. In this study, because ON yields were calculated on a short period (max. 1 hour), wall losses at this time scale are estimated to be less than 10%. This is confirmed by the good linearity of the plots.





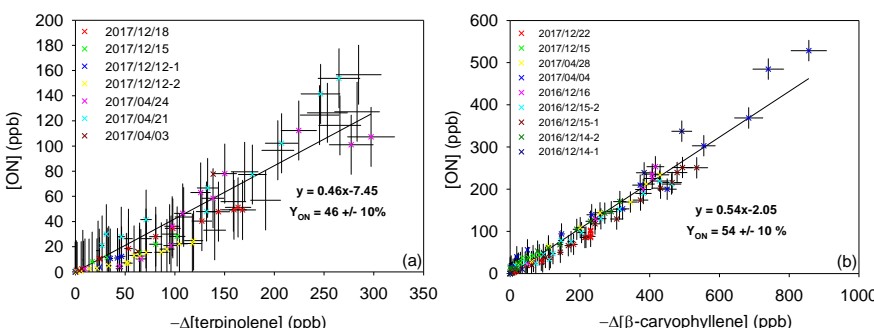

**Figure 6: Gas phase organic nitrates production *vs* loss of terpinolene (a) and for βaryophyllene (b).**

Molar $Y_{ONg}$ were found to be $47 \pm 10$ % for terpinolene and $43 \pm 10$ % for β-caryophyllene. These yields are in good agreement with previous studies performed for other BVOCs and showing that ONs are major products of

BVOC + $NO_3$ reactions. ON yields measured for isoprene and monoterpenes are indeed higher than 30 %. For example, limonene ON yield vary between 30 and 72 % (Fry et al., 2014; Hallquist et al., 1999; Spittler et al., 2006) and β-pinene between 40 and 74 % (Fry et al., 2014; Hallquist et al., 1999; Boyd et al., 2015). For α- and γ-terpinene (Fouqueau et al., 2020a) very close yields (47 and 44 % respectively) have been measured. The only exception is α-pinene, for which yields vary between 10 and 30 % (Fry et al., 2014; Hallquist et al., 1999; Spittler

et al., 2006). Its main product is indeed an aldehyde, with a high vapor pressure, that is not contributing to SOA phase.

Organic nitrates may partition between gas and aerosol phases. Hence, yields of total organic nitrates in the particle phase ($Y_{ONp}$) have been determined using FTIR analyses of the collected filters. Results are shown in Table 3. For terpinolene, molar yields range between 7 and 23%. The variability of these yields can be explained by the fact

that, as SOA yields, they depend on the reacted BVOC concentration. Indeed, for high concentration experiments (~350 ppb), yields are around 20%, whereas they are around 7% for the low concentration experiments (~180 ppb). For β-caryophyllene, $Y_{ONp}$ range between 21 and 25%, and appears to be less subject to variability.

In order to evaluate the fraction of organic nitrates in SOA, $Y_{ONp}$ have been compared to SOA yields. For this comparison, both yields have to be expressed in mass. To do so, a unique molecular weight which is representative

of the expected oxidation products has been considered: for terpinolene, a hydroxynitrate ($C_{10}H_{17}O_4N$) having a molecular weight of 215 g mol$^{-1}$ has been chosen. For β-caryophyllene, the same type of compound has been chosen ($C_{15}H_{25}NO_3$), with a molecular weight of 283 g mol$^{-1}$. Both compounds were detected as oxidation product by PTR-ToF-MS. It is clear that this assumption generates a large error on ON mass yield, particularly if other products are formed with higher molecular weights (e.g. by polymerization in condensed phase). Nevertheless, as

oxidation products were not quantified individually, this method is the only way to estimate the contribution of ONs to the aerosol phase. The ratio $Y_{ONp,\ mass}$ / $YS_{OA,mass}$ are also shown in Table 3. From these results, it is estimated that organic nitrates represent ~50 % of the SOA for terpinolene and ~80 % for β-caryophyllene and are therefore major components of the SOA produced by BVOC + $NO_3$ reaction. It should be noticed that if higher





molecular weight products were formed, these ratios would be even greater. The value obtained for β-caryophyllene is in very good agreement with the ratio of 80% provided by Fry et al., 2014. These results are also in good agreement with fields studies (Kiendler-Scharr et al., 2016; Ng et al., 2017) which have observed that organic nitrates are major component of organic aerosols, with a proportion that can reach almost 80 %. Even if organic nitrates can be produced by other reactions, an enhancement of organic nitrates in SOA has been observed by several studies in regions impacted by $NO_3$ radical during the night (Gómez-González et al., 2008; Hao et al., 2014; Iinuma et al., 2007) and also in forest regions affected by urban air masses (Hao et al., 2014). This result thus confirms both the importance of $NO_3$ chemistry in SOA formation and the major contribution of organic nitrates in SOA formation.

### 4.3 Products at molecular scale and mechanisms

To propose explanations for the measured yields, mechanisms have been build, using the molecular scale PTR-ToF-MS identification of gas phase products. By using two ionization modes, (i.e. $H_3O^+$ and $NO^+$), an accurate identification of the molecules was possible. Detected signals in both ionization modes and corresponding raw formula are summarized in Table 4. Products with molecular weights of 58, 142 and 168 g mol$^{-1}$ for terpinolene have been detected with high intensities. For β-caryophyllene, main signals were measured for products having molecular weights of 221 and 236 g mol$^{-1}$. Many of the products which were detected are nitrogenous species which is in good agreement with the measurement of high organic nitrates yields. Mechanisms have been proposed in Fig. 8 for terpinolene and in Fig. 9 for β-caryophyllene. Time profiles of PTR-ToF-MS signals (see Fig. S1) were also used to determine whether the products are primary or secondary ones. First generation products are framed in blue and second generation ones in pink.

**Table 4: Products detected for Terpinolene (a) and β-caryophyllene (b) with PTR-ToF-MS $H_3O^+$ and $NO^+$ ionization modes: formula and molar masses, detected masses, ionization processes ($H^+$: proton adduct, $NO^+$: $NO^+$ adduct, CT: charge transfer and PL: proton loss), peak intensity, and comportments.**

| | Molecule | | $H_3O^+$ ionization mode | | | | $NO^+$ ionization mode | | | |
|---|---|---|---|---|---|---|---|---|---|---|
| | Raw formula | M (g/mol) | m/z | Process | Intensity | Behavior | m/z | Process | Intensity | Behavior |
| (a) | $C_3H_6O$ | 58 | 59.0579 | $H^+$ | +++ | Primary | 58.0411 | CT | ++ | Primary |
| | $C_7H_{10}O$ | 110 | 111.0842 | $H^+$ | ++ | Primary | 110.0753 | CT | ++ | Primary |
| | $C_7H_{10}O_{22}$ | 126 | 127.0642 | $H^+$ | ++ | Primary | 126.0584 | CT | + | Primary |
| | $C_7H_{10}O_3$ | 142 | 143.0581 | $H^+$ | ++ | Primary | 142.0539 | CT | +++ | Primary |
| | $C_{10}H_{16}O$ | 152 | 153.1171 | $H^+$ | ++ | Secondary | / | / | / | / |
| | $C_{10}H_{16}O_2$ | 168 | 169.0952 | $H^+$ | + | Primary | 168.1038 | CT | +++ | Primary |
| | $C_{10}H_{16}O_3$ | 184 | 185.0877 | $H^+$ | ++ | Primary | / | / | / | / |
| | $C_8H_{17}N_2O_4$ | 205 | / | / | / | / | 205.17 | CT | + | Secondary |
| | $C_{10}H_{15}NO_4$ | 213 | 214.1262 | $H^+$ | + | Primary | / | / | / | / |
| | $C_{10}H_{17}NO_4$ | 215 | 216.0816 | $H^+$ | + | Primary | 214.1174 | PL | + | Primary |

|  |  |  |  |  |  | 245.1796 | NO⁺ | + | Primary |
|---|---|---|---|---|---|---|---|---|---|
|  | $C_{10}H_{15}NO_5$ | 229 | 230.0836 | H⁺ | ++ | Primary | 229.1056 | CT | + | Primary |
|  | $C_9H_{15}NO_6$ | 233 | 234.0635 | H⁺ | + | Detected | 233.0877 | CT | + | Primary |
|  | $C_{10}H_{15}NO_6$ | 245 | 246.0999 | H⁺ | + | Primary | / | / | / | / |
|  | $C_{10}H_{15}NO_7$ | 261 | 262.1143 | H⁺ | + | Detected | / | / | / | / |
|  | $C_{10}H_{16}N_2O_7$ | 276 | / | / | / | / | 276.1304 | CT | + | Detected |
|  | $C_8H_{18}N_2O_9$ | 286 | 287.1717 | H⁺ | + | Detected | / | / | / | / |
|  | $C_{10}H_{14}O_{10}$ | 294 | 295.0404 | H⁺ | + | Detected | 293.0629 | PL | + | Detected |
| (b) | $C_{15}H_{24}O$ | 220 | 221.1674 | H⁺ | +++ | Primary | 220,2033 | CT | ++ | Primary |
|  | $C_{14}H_{22}O_2$ | 222 | 223.1717 | H⁺ | + | Secondary | 222,1779 | CT | + | Secondary |
|  | $C_{15}H_{24}O_2$ | 236 | 237.2017 | H⁺ | +++ | Primary | 236,202 | CT | +++ | Primary |
|  | $C_{14}H_{22}O_3$ | 238 | 239.2183 | H⁺ | + | Secondary | / | / | / | / |
|  | $C_{15}H_{24}O_3$ | 252 | 253.1838 | H⁺ | + | Secondary | 252,2279 | CT | + | Detected |
|  | $C_{14}H_{22}NO_4$ | 267 | 268.2034 | H⁺ | + | Detected | 267,3576 | CT | + | Detected |
|  | $C_{15}H_{23}NO_3$ | 281 | 282.1562 | H⁺ | ++ | Primary | 311,2733 | NO⁺ | + | Detected |
|  | $C_{15}H_{25}NO_3$ | 283 | 284.2149 | H⁺ | + | Primary | 282,4786 | PL | ++ | Primary |
|  | $C_{15}H_{25}NO_5$ | 298 | 299.1841 | H⁺ | + | Primary | 298,2353 | CT | + | Primary |
|  | $C_{15}H_{23}NO_6$ | 313 | 314.2904 | H⁺ | + | Detected | / | / | / | / |
|  | $C_{15}H_{23}N_2O_6$ | 327 | / | / | / | / | 327,1502 | CT | + | Detected |
|  | $C_{15}H_{24}N_2O_7$ | 344 | 345.2471 | H⁺ | + | Detected | / | / | / | / |

In addition, for experiments on terpinolene, acetone was detected by FTIR and its formation yield has been measured. Fig. 7 shows the concentration of acetone plotted against the consumption of terpinolene. Every experiments shows similar and linear tendencies, within uncertainties. Acetone appears to be a primary product, with a production yield of 23 ± 5 %. Terpinolene is thus a major precursor of acetone.

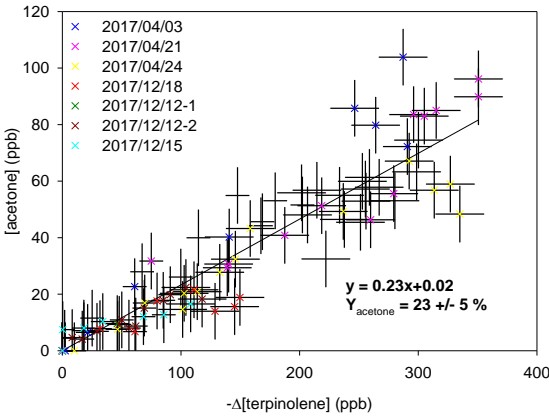

**Figure 7: Acetone production *vs* loss of terpinolene.**





### 4.3.1 Terpinolene oxidation scheme

NO$_3$ radical reacts with terpinolene by addition onto one of the two double bonds (H-atom abstraction is considered
negligible), each addition leading to the formation of two possible nitrooxy alkyl radicals. According to the SAR
developed by Kerdouci et al., 2014, the exocyclic double bond is expected to be 5 times more reactive than the
endocyclic one as it is more substituted. Nevertheless, all possible pathways were considered here but the pathways
for only two radicals are presented in Fig. 8 in order to facilitate the reading (see Fig. S2 for the two others). In
most cases, the products formed are isomers and cannot be distinguished from one path to another with the
techniques used here.







**Figure 8: Proposed mechanism for terpinolene. First generation products are squared in blue and second generation ones in red. Alkoxy fragmentation products are squared according to the location of the fragmentation. Molecular weight, vapor pressures and the gas/particle partition are shown next to the molecules.**

Nitrooxy alkyl radicals can then react with $O_2$ to form a peroxy radical ($RO_2$) via the reaction 2. The formation of an epoxide has also been observed (152 g mol$^{-1}$, reaction 3), using both $NO^+$ (m/z 152) and $H_3O^+$ (m/z 153)


ionization modes. $RO_2$ radicals then react following different pathways: they can react with $NO_2$ to form a peroxynitrate, $RO_2NO_2$ (MW = 276 g mol$^{-1}$) following reaction 4. It was detected in $NO^+$ ionization mode at m/z 276. This reaction is usually negligible in the atmosphere, but can be significant in simulation chambers due to high $NO_2$ concentrations. It can also react with another peroxy radical ($RO_2 + RO_2$, reaction 5) to form a characteristic hydroxynitrate (MW = 215 g mol$^{-1}$) and a ketonitrate (MW = 213 g mol$^{-1}$). Both were detected at m/z 216 (M+1) in $H_3O^+$ ionization mode and m/z 214 (M-1) in $NO^+$ mode for the hydroxynitrate, and at m/z 214 (M+1) in $H_3O^+$ mode and 243 (M+30) in mode $NO^+$ mode for the ketonitrate. This reaction involving an H-atom transfer is possible only if the carbon that carries the peroxy radical group is linked to a Hydrogen, i.e. for primary and secondary peroxy radicals. Here, this reaction is thus possible only for the peroxy radical coming from the addition on the endocyclic double bond shown in Fig. 8. Finally, peroxy radical can react with another $RO_2$ or with $NO_3$ radical (reactions 6 and 6' respectively) to form an alkoxy radical (RO).

RO radicals can then evolve following reactions 7, 8 and 9. They can react with $O_2$ (reaction 7) to form the same ketonitrate as the one formed by reaction 5 (MW = 213 g mol$^{-1}$). In case of $NO_3$ addition onto the endocyclic double bond, the resulting alkoxy radical can decompose following reaction 8, leading to the formation of an alkyl radical, which then reacts following previously mentioned pathways to form a diketonitrate (MW = 229 g mol$^{-1}$, framed in green in Fig. 8). This trifunctional product has been detected both in $H_3O^+$ (m/z 230) and $NO^+$ ionization modes (m/z 229). This alkoxy radical can also decompose by a scission of the C(ONO$_2$)-CH(O$^\bullet$) bond (reaction 9), leading to the formation of a dicarbonyl ring opening product of molecular weight MW = 168 g mol$^{-1}$ (detected at m/z 169 in $H_3O^+$ mode and m/z 168 in $NO^+$ mode). In case of $NO_3$ addition onto the exocyclic double bond, the resulting alkoxy can decompose to form a carbonyl product of molecular weight MW = 110 g mol$^{-1}$ (detected at m/z 111 in $H_3O^+$ mode and m/z 110 in $NO^+$ mode) and acetone. Acetone has been detected with a formation yield of 23%. Considering that this pathway is the only one allowing the primary production of acetone, a tentative determination of branching ratio has been made. As mentioned previously, $NO_3$ addition to the exocyclic double bond is expected to be the major pathway. The two resulting alkoxy radicals (see Figure S3) can both produce acetone by decomposition but with expected different yields. The radical shown in Figure 8, i.e. the one having the radical group on the isopropyl group, is expected to produce mainly acetone whereas the other one shown in Figure S2, i.e. the one having the radical group on the cycle, is expected to decompose following the three possible pathways which have very close activation energies (Vereecken and Peeters, 2009). Thus, considering the same probability for the three decomposition pathways, acetone production yield would be around 30%. Experimental acetone yield being 23%, this would suggest that the alkoxy having the radical group on the cycle is predominant.

Primary products can themselves react with $NO_3$ because they still possess a double bond, leading to the formation of second-generation products, squared in red in Fig. 8. Second-generation products coming from the carbonyl and the dicarbonyl products have been identified: a tri-carbonyl compound (MW = 142 g mol$^{-1}$) and two epoxides (MW = 184 g mol$^{-1}$ and MW = 126 g mol$^{-1}$).

Calculated vapor pressures and their estimated partition in the SOA are shown next to the products in Fig. 8. Among the first generation products, two are likely to participate to SOA formation: the hydroxynitrate and the diketonitrate. The hydroxynitrate is a product characteristic of the $RO_2 + RO_2$ pathway and has a low vapor pressure because of presence of hydrogen bonds. This compound is estimated to be at 40 % in SOA phase. However, considering that the addition of $NO_3$ proceeds mainly by addition onto the exocyclic double bond, the





formation of the hydroxynitrate is expected to be minor. The diketonitrate (MW = 229 g mol-1) is also expected to significantly contribute to SOA formation with a partition of 90 % in SOA phase. It can be formed by both additions of $NO_3$ onto the exocyclic and endo cyclic C=C bonds. For this trifunctional product, the associate partitioning coefficient, $K_p$ has been calculated following Eq. (10). Considering the uncertainty on $\xi^i_{aer}$ due to the vapor pressure estimation, it can vary from $1.1 \times 10^{-2}$ to $3.4 \times 10^{-3}$ $m^3$ $\mu g^{-1}$. This value is consistent with the partitioning coefficient found with the two product model from Eq. (8) ($K_{p,1}$ = 6.7 $10^{-3}$ $m^3$ $\mu g^{-1}$), within the associated estimated uncertainty on $K_p$.

Identified secondary products have high vapor pressures and thus may not contribute to the SOA formation. Other products with molecular weights close to 290 g mol$^{-1}$ have been detected with weak signals but were not identified. Due to their high molecular weights, they could significantly contribute to SOA. In addition, other secondary products may be formed without being detected by PTR-ToF-MS due to their too low volatility.

### 4.3.2 β-caryophyllene oxidation scheme

β-caryophyllene has two double bonds, one exocyclic and one endocyclic, but according to Kerdouci et al., 2014 SAR, the exocyclic bond is expected to be approx. 40 times less reactive than the endocyclic one CH=C< one because it is less substituted. So only the addition onto the endocyclic bond has been considered here, leading to the formation of two possible nitrooxy alkyl radicals (see Fig. 9).



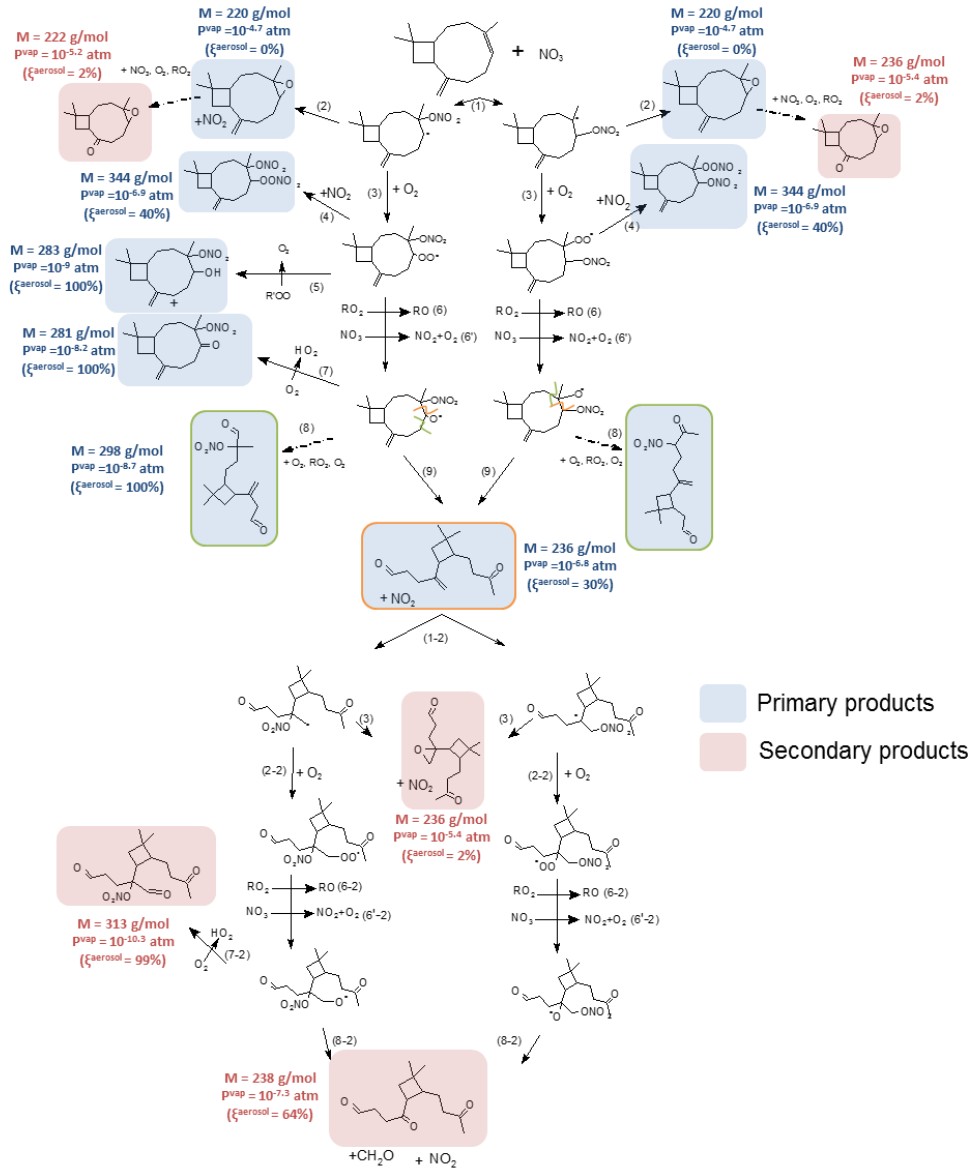


**Figure 9: Proposed mechanism for β-caryophyllene. First generation products are squared in blue and second generation ones in red. Alkoxy fragmentation products are squared according to the location of the fragmentation. Molecular weight, vapor pressures and the gas/particle partition are shown next to the molecules.**

Like for terpinolene, alkyl radicals can evolve following two pathways: i) the formation of an epoxide (MW = 220 g mol$^{-1}$, reaction 2), detected at m/z 221 in H$_3$O$^+$ ionization mode and m/z 220 in NO$^+$ mode; ii) the formation of a peroxy radical, by reaction with O$_2$ (reaction 3). Under high NO$_2$ levels, RO$_2$ radicals can then react with NO$_2$ to



form peroxynitrates (reaction 4) of molecular weight MW = 344 g mol$^{-1}$ (identified at m/z 345 in H$_3$O$^+$ mode and m/z 344 in NO$^+$ mode). RO$_2$ radicals can also undergo self-reactions leading to the formation of a characteristic

hydroxynitrate and a ketonitrate (reaction 5) of molecular weights MW = 283 and 281 g mol$^{-1}$ respectively. The hydroxynitrate has been detected at m/z 284 in H$_3$O$^+$ ionization mode and m/z 282 in NO$^+$ mode and the ketonitrate at m/z 282 (M+1, H$_3$O$^+$) and m/z 311 (M+30, NO$^+$). As mentioned previously, this reaction is only possible when the carbon atom which carries the peroxy group is linked to an H-atom, so here, only for one of the two peroxy radicals. Finally, RO$_2$ radicals can react with another peroxy radical or with NO$_3$ (reactions 6 ad 6') to form an

alkoxy radical. This last one can react with O$_2$ to form a ketonitrate (MW = 281 g mol$^{-1}$). The alkoxy radicals can decompose by a scission of the C(ONO$_2$)-CH(O•) bond associated with a loss of NO$_2$, to form a dicarbonyl product of MW = 236 g mol$^{-1}$ (m/z 237 in H$_3$O$^+$ mode and m/z 236 NO$^+$ mode, reaction 9, squared in green). It can also decompose by a C-C breaking on the other side of the alkoxyl group (reaction 8) to form a trifunctional compound (MW = 298 g mol$^{-1}$), detected at m/z 299 (H$_3$O$^+$) and m/z 298 (NO$^+$). It should be noticed that, in the case NO$_3$

radical adds onto the exocyclic double bond, formaldehyde is expected to be formed (see Fig. S3) but it was not detected with FTIR (with a detection limit close to 10 ppb). This information confirms that this pathway is minor.

Even though the reaction of NO$_3$ on the remaining CH$_2$=C< double bond is expected to be slow, secondary products have been detected and shown in red in Fig. 9. Second generation epoxides (MW = 252 g mol$^{-1}$, m/z 253 in H$_3$O$^+$ mode and m/z 252 NO$^+$ mode and MW = 222 g mol$^{-1}$, m/z 223 in H$_3$O$^+$ mode and m/z 222 in NO$^+$ mode) have been

measured (reaction 3). Also a carbonyl compound of MW = 238 g mol$^{-1}$ (m/z 239 in H$_3$O$^+$ mode and m/z 238 in NO$^+$ mode) coming from the decomposition of the alkoxy radical (reaction 9-2) has been detected. A trifunctional species can also be formed by the reaction of the alkoxy radicals with O$_2$ (reaction 8-2). This diketonitrate (MW = 313 g mol$^{-1}$) was detected in H$_3$O$^+$ mode (m/z 314). Finally, a nitrogen product which can be a dinitrate has been detected at m/z 327 in NO$^+$ ionization mode, but has not been identified.

As for terpinolene, estimated vapor pressures of detected products and corresponding partitioning ratio between the gaz and aerosol phase are shown next to the products in Fig. 9. Because β-caryophyllene is a sesquiterpene (C15), most of the oxidation products have very low vapor pressures and can thus contribute to SOA formation. This is in good agreement with the high SOA yields observed even for low aerosol mass loading. Only few products formed by fragmentation processes have relatively high volatility, thus explaining SOA yields below

100%. Finally, many identified products are also organic nitrates, in good agreement with gas phase observations.

## 5 Discussion & comparison

Yields measured in this study are summarized in Table 5 and compared to those obtained for other BVOCs by

previous studies. It can be observed that oxidation of terpinolene and β-caryophyllene produces large amounts of SOA and ONs, similarly to other BVOCs, with two notable exceptions for α-pinene and α-terpinene. In the case of α-pinene, larger formation yield of carbonyls was observed in comparison to the others BVOCs (Ng et al. 2017). These carbonyl compounds being more volatile than ONs, several previous studies suggest that there is a correlation between ONs and SOA yields (Hallquist et al. 1999; Fry et al. 2014). Indeed, α-pinene has a low

organic nitrate yield, corresponding to almost no SOA production, when limonene and Δ-carene both exhibit high SOA and organic nitrate yields. However, the results obtained in a previous comparative study (Fouqueau et al,



2020b) for α- and γ-terpinene show that α-terpinene does not follow this correlation, as it produces large amount of organic nitrates but almost no SOAs. To interpret these observations, the mechanisms have to be considered.

**Table 5: Mean SOA and organic nitrate yields obtained in this study for terpinolene and β-caryophyllene and for other terpenes in the literature.**

| Compound | Formula | $Y_{SOA}$ (10 µg m$^{-3}$) | $Y_{ON, total}$ | $Y_{ONp, mass}$ / $Y_{SOA,mass}$ | Ref. |
|---|---|---|---|---|---|
| Terpinolene | | 5 % | 69 ± 24% | 28-66% | This study |
| β-caryophyllene | | 40 % | 79 ± 23% | 79% | This study |
| γ-terpinene | | 10 % | 55 ± 15% | 7-50% | Fouqueau et al., 2020a |
| α-terpinene | | 1.2 % | 48 ± 12% | 86-125% | Fouqueau et al., 2020a |
| Isoprene | | 12 % | 62-78% | / | Ng et al., 2008 : Rollins et al., 2009 |
| α-pinene | | 0% | 10% | / | Fry et al., 2014 |
| β-pinene | | 33-44% | 45-74% | / | Boyd et al., 2015; Fry et al., 2014 |
| Δ-carene | | 38-65% | 77% | / | Fry et al., 2014 |
| limonene | | 44-57% | 77% | / | Fry et al., 2014 |

As discussed previously in section 4.3, but also in Fouqueau et al., 2020a, two mechanism steps are critical for the SOA formation: the peroxy and the alkoxy reaction pathways. For the peroxy radicals, this study has shown the 610 hydroxynitrates coming from the reaction $RO_2 + RO_2 \rightarrow ROH + R(O)$ have low vapor pressures and can contribute to SOA formation. In the case of terpinolene, this reaction is less favorable than for β-terpinene for example, because the reaction is estimated to proceed mainly by addition of $NO_3$ onto the fully substituted exocyclic double bond, leading to tertiary peroxy radicals. Even though these hydroxynitrates were detected, their formation yields should be low. For the alkoxy radicals, several decomposition pathways can occur, forming



different types of products having different volatilities: the scission of the C(ONO$_2$)-CH(O$^\bullet$) bond leads to the formation of volatile dicarbonyl products. On contrary, when the alkoxy decomposes by a scission of the C-C bond located on the other side of the alkoxy group, it produces a keto-nitrooxy alkyl radical which then evolves to form a low vapor trifunctional species (diketonitrate). The major role of this two steps has already been pointed out by previous studies. The role of RO$_2$ + RO$_2$ reaction has been shown to play a significant role in the SOA formation from isoprene (Ng et al., 2008). The role of the alkoxy radical decomposition has already been raised by Kurten et al., 2017 suggesting that for Δ-carene, which has a high SOA yield, the decomposition of alkoxy radicals can lead to the formation of keto-nitrooxy-alkyl radicals, whereas for α-pinene, the alkoxy radicals decompose almost exclusively to form the dicarbonyl compound, explaining the low SOA and ON yields. The mechanisms of terpinolene are thus in good agreement with these previous studies.

SOA yields obtained for β-caryophyllene are very high and this can easily be explained by the size of this precursor (C$_{15}$). β-caryophyllene is the only sesquiterpene for which data have been provided and comparison of its SOA yield with those obtained for terpenes is not fully relevant. Nevertheless, the same key steps have been noticed in the mechanism. The addition of NO$_3$ onto the endocyclic double bond is expected to be the major pathway leading to the formation of the same types of functionalized products as those observed for terpinolene (hydroxynitrates, ketonitrates, diketonitrates), but here having much lower vapor pressures.

Organic nitrate yields of both studied compounds are around 50 %. They can be compared to those measured for other BVOCs, presented in Table 5: within the uncertainties, they appear to be similar to those of α- and γ-terpinene (48 and 55 % respectively, Fouqueau et al., 2020a). Limonene has yield between 30 and 72 % (Hallquist et al., 1999; Spittler et al., 2006) and β-pinene, between 22 and 74% (Boyd et al., 2015; Fry et al., 2014; Hallquist et al., 1999). They also appear similar to those of Δ-carene (68-77%, Fry et al., 2014; Hallquist et al., 1999) and isoprene (62-78 %, Rollins et al., 2009) within the uncertainties. BVOC+NO$_3$ reaction are therefore major sources of ONs.

For both compounds, epoxides have been detected. They were not quantified, but based on previous studies, their formation yields are expected to be low. Their formation is considered favored only at low oxygen concentration (Berndt and Böge, 1995). Even if, their detection is rare in previous studies, their formation was already observed in the same experimental conditions in Fouqueau et al., 2020a. They were also measured by Skov et al., 1994, which studied the oxidation of some alkenes and isoprene by NO$_3$. Low epoxides yields have also been reported by Wangberg et al., 1997 (3 % for α-pinene) and Ng et al., 2008 (>1 % for isoprene).

**6 Conclusions and atmospheric impacts**

In summary, this study has provided kinetic and mechanistic data on the reaction between nitrate radicals and two BVOCs, terpinolene and β-caryophyllene. For the first time, an absolute rate determination was conducted for β-caryophyllene. Both compounds have been studied using relative and absolute rate determinations leading to kinetic data in good agreement. Due to the presence of two double bonds, they appear to be very reactive towards nitrate radical. As far as we know, this is also the first mechanistic study of terpinolene + NO$_3$ reaction, and the first determination of ON yields for β-caryophyllene. They both produce large amounts of ONs in gas phase, with yields around 50 %. These compounds have been also detected in particle phase, with production yield of 25 % for the two compounds. In total, these reactions produce around 70-80 % of organic nitrates. These compounds





were shown to be also good SOA precursors. At 10 µg m$^{-3}$, terpinolene has an SOA yield of 5 %, when β-caryophyllene has a yield of 40 %. The last one produces high amount of SOA, even for low aerosol mass loading.

For both compounds, SOA formation has been explained thanks to the detection of oxidation products at the molecular scale that allowed proposing mechanisms. The SOA yield of terpinolene can be explained by the formation of two types of low volatility products: a trifunctional species and a hydroxynitrate. High SOA yields observed for β-caryophyllene can be explained by the formation of several high molecular weight products. For both compounds, preferential pathways have been proposed.

In order to evaluate the contribution of the NO$_3$-initiated oxidation to the total degradation of these BVOCs, atmospheric lifetimes have been estimated using NO$_3$ concentrations of 10 ppt (typical nighttime concentration) and 0.1 ppt (low insolation diurnal concentration, (Khan et al., 2015). It should be noticed that terpinolene is intensively emitted during both day and night (Lindwall et al., 2015). These lifetimes are compared to those estimated for OH and ozone oxidation in Table 6. It can be observed that terpinolene and β-caryophyllene have

very short lifetimes (few minutes) towards NO$_3$ radical in nighttime conditions, confirming that NO$_3$ oxidation is a major sink for these compounds. During the day, in low sunlight conditions, lifetimes are still short (between 2 and 7 hours). They nevertheless are longer than those estimated for OH and ozone chemistries. NO$_3$ is thus a minor oxidant under these diurnal conditions. These short lifetimes also demonstrate that oxidation products will be formed close to the emission area.

**Table 6: Atmospheric lifetimes of terpinolene and β-caryophyllene with respect to their oxidation by NO$_3$ and OH radicals and by ozone.**

| Compound | $\tau_{NO3}$ * | $\tau_{NO3}$** | $\tau_{OH}$*** | $\tau_{O3}$*** |
|---|---|---|---|---|
| | (min) | | | |
| terpinolene | 1,3 | 131 | 38[2] | 15[1] |
| β-caryophyllene | 4,2 | 434 | 42[3] | 2[1] |

* calculated with [NO$_3$] = $2.5 \times 10^8$ molecule cm$^{-3}$ (10 ppt)

** calculated with [NO$_3$] = $2.5 \times 10^6$ molecule cm$^{-3}$ (0.1 ppt)

***calculated with [OH] = $2 \times 10^6$ molecule cm$^{-3}$ and [O$_3$] = $7 \times 10^{11}$ molecule cm$^{-3}$

[1] calculated with rate constant recommended by IUPAC

[2] calculated with rate constant from Corchnoy and Atkinson, 1990

[3] calculated with rate constant from Shu and Atkinson, 1995

One characteristic feature of the oxidation of BVOCs by NO$_3$ radical is that it produces large amount of organic

nitrates in both gas and aerosol phases. Even though OH-initiated oxidation can also produce organic nitrates (through RO$_2$ + NO reactions), yields are usually lower (Lee et al. 2006). Another major finding of this study is that the NO$_3$-oxidation of β-caryophyllene, and to a lesser extend of terpinolene, produces large amounts of SOA. The yields obtained in this study can be compared to those measured in previous studies for ozonolysis and OH oxidation. First, concerning the oxidation by OH radicals, SOA yields measured for terpinolene were shown to be

close to those measured for NO$_3$ oxidation: for low NOx conditions, the SOA yield was found to be around 3% at M$_0$ = 10 µg m$^{-3}$ but that can reach 40 % for higher aerosol mass loadings (Friedman and Farmer, 2018; Lee et al.,



2006b). For β-caryophyllene, the SOA yields were shown to reach 68% (Lee et al., 2006). Regarding the ozonolysis, SOA yields have been found to be 20 % for terpinolene and 45 % for β-caryophyllene (Lee et al., 2006). Regarding these results, the oxidation by $NO_3$ produces similar amounts of SOA than the other oxidants.

However, the chemical composition of the aerosol phase is significantly different.

In conclusion, the most important impacts of this chemistry rely on the formation of large amounts of organic nitrates (present in both gas and aerosol phases) and SOA. Organic nitrates play a key role in tropospheric chemistry because they behave as NOx reservoirs, carrying reactive nitrogen in remote areas. Their chemistry in gas and aerosol phases is nevertheless still not well documented. Considering that our study shows a large

production of multifunctional organic nitrates, it is necessary to better understand their reactivity in order to better evaluate their impacts. Formation of SOA seems on the other hand, to be strongly dependent of the structure of the BVOC. Studies at molecular scale are thus crucial to better evaluate the impact of this chemistry on the SOA formation.

*Data availability:* SOA yields and rate constant for the $NO_3$ oxidation of terpinolene and β-caryophyllene are available Table 2. It is also available through the Library of Advanced Data Products (LADP) of the EUROCHAMP data center (https://data.eurochamp.org/data-access/ gas phase-rate-constants/, last access: 01 May 2021, Fouqueau et al., 2020c). The simulation chamber experiments raw data which have served as basis for both the kinetic and mechanistic work are available through the Database of Atmospheric Simulation Chamber Studies

(DASCS) of the EUROCHAMP data center (https://data.eurochamp.org/data-access/chamber-experiments/, last access: 01 May 2021, Fouqueau et al., 2021d).

*Author contributions:* BPV and MCi coordinated the research project. AF, BPV, MCi and JFD designed the experiments in the simulation chambers. AF performed the experiments with the technical support of MC and EP

and performed the data treatment and interpretation with MCi and BPV. AF, BPV and MCi wrote the paper, and AF was responsible for the final version of the paper. All coauthors revised the content of the original manuscript and approved the final version of the paper.

*Competing interests.* The authors declare that they have no conflict of interest.

*Acknowledgements.* The authors thank Marie-Thérèse and Jean-Claude Rayez (ISM, Bordeaux, France) for helping understanding the reactivity with theoretical calculation and Marie Camredon (LISA, Créteil, France) for helping with the GECKO-A website. The authors also gratefully acknowledge CNRS-INSU for supporting CESAM National Facility as a component of the ACTRIS French Research Instructure. The AERIS data center (https://www.aeris-data.fr/) is also gratefully acknowledged for curing and distributing the data as the datacenter

of the EUROCHAMP-2020 Integrated Activities and, in the future, as a pillar of ACTRIS ERIC.

*Financial support.* This work was supported by the French national programme LEFE/INSU (CNRS) and by the Horizon 2020 Research and Innovation Program through the EUROCHAMP-2020 Infrastructure Activity under grant agreement no. 730997. This work was also supported by grants from Région Ile de France.

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
