# Peer review of "An experimental study of the reactivity of terpinolene and $\beta$ caryophyllene with the nitrate radical"

_Atmospheric Chemistry and Physics, 2021_

## Author Response (AR1)

First of all, the authors would like to thank the anonymous referee for this discussion and its constructive comments, corrections and suggestions that ensued. We have carefully replied to all its comments and the paper has been improved following its recommendations. Answers have also been provided for all comments and changes have been performed accordingly. Please find below the answers to the comments:

**General comments:**

**Yields of SOA in chamber experiments such as these are generally challenging, with various complications such as: a dependence on the starting concentration of the alkene reactant; a dependence on the amount of alkene reactant that has been consumed; and a dependence on the aerosol mass loading. Given that chamber experiments tend to be run under higher than ambient concentrations of the alkenes and their oxidation products, it is generally unclear whether the yields that are presented here would actually apply to ambient conditions.**

**Let us assume that if one were to oxidize enough of almost anything in a chamber environment, it would eventually form aerosol as the chamber contents reach supersaturation with respect to a condensable component. For example, some of the estimated vapour pressures that are present in the beta-caryophyllene mechanism are very low (~5 × 10-11 atmospheres). I think that this would mean that a concentration of such a compound would reach supersaturation in the air at a concentration of approximately 50 ppt. If we took a typical concentration of consumed [BVOC] to be 400 ppb, then we could estimate that if the branching ratio towards this low vapour pressure product was any larger than 1/8000, then such a molecule would overcome its vapour pressure within your chamber conditions, and would therefore begin to condense.**

**So the question remains, what could we actually expect to be the true SOA yield to be under ambient conditions, considering that BVOCs would be diluting and mixing during oxidation, in contrast to the chamber experiments presented here? Perhaps it is mostly the different concentration regimes that are used in the various literature experiments that lead to the spread of SOA yields?**

We agree that performing experiments using BVOC mixing ratios of few hundreds ppb may lead to an overestimation of the SOA yield, as even very minor oxidation products may reach the saturating vapor pressure, while it may not be the case in ambient air. However, investigating mechanisms requires being able to detect oxidation products. Performing experiments with BVOC mixing ratios of few tens-hundreds ppt (which is representative of ambient air) would allow to observe under more realistic conditions whether SOA is formed, but certainly not to identify the species which are responsible for the SOA formation and to understand the chemical mechanisms. The objective of the mechanistic study here is to understand the chemical processes involved in the SOA formation. Therefore, we consider that there is no perfect experiment to investigate together the SOA formation under fully representative ambient air conditions and the chemical mechanisms.

However, to limit the risk of SOA yield overestimation, the BVOC was slowly oxidized (within almost one hour for some experiments) by slowly injecting $N_2O_5$ into the chamber. This method also allowed to observe SOA formation under low aerosol mass contents (few tens $\mu g/m^3$ at the beginning of the oxidation) which are representative of mixed biogenic/anthropogenic environments.

In addition, the SOA yield plots were fitted by a two-products parametrization which has precisely the objective to take into account the dependence of the SOA yield with the aerosol concentrations. For terpinolene, the fit was very good, even for low aerosol content. It suggests that the particle phase products can be simulated with only one family of same volatility (Kp = $6.7 \times 10^{-3}$) which is formed with

a high stoichiometric coefficient ($\alpha$ = 0.6) suggesting that these products are not minor ones. In addition, the obtained partitioning coefficient is in agreement with those calculated for the detected oxidation products. These results make that we are confident with the possibility of extrapolating these results to ambient conditions. For $\beta$-caryophyllene, the experimental results are more scattered and the model is less well constraint for low aerosol mass loading. So results are more uncertain.

Finally, maybe more than the dependence to the BVOC concentration or to the aerosol content, our study has demonstrated that the chemical regime is a key aspect for the SOA formation. In particular, the peroxy radical chemistry ($RO_2+RO_2$, $RO_2+NO_3$, …) has been shown to play a key role in the SOA formation and these processes are dependent on mixing ratios of $RO_2$, $NO_3$ or $HO_2$ in the air mass. So to better represent the SOA formation in models, we really need to better understand these mechanisms and their impact on aerosol formation.

**Added to this, the authors are quite inconsistent in the SOA yields that they mention in the abstract and throughout the text. In the abstract, they list maximum SOA yields of 60% for both compounds. In section 4.1, they suggest that beta-caryophyllene could be as high as 90%. In the conclusions section, the authors seem to favour the 5% and 40% values at 10 micrograms per cubic metre. I am therefore not exactly sure which of these numbers (if any) I should be considering from this work.**

The referee is right pointing out the inconsistency between maximal SOA yields in the abstract and in the text. The sentence has been modified in the abstract (L. 24)**:** "**around 60 % for terpinolene and 90 % for β-caryophyllene.**" In the conclusion section, the value indicated is the one obtained for an aerosol mass loading of 10 µg m$^{-3}$ in order to allow for comparison with previous studies (Fry et al., 2014 or Ng et al., 2017).

**Minor comments:**

**Line 13: … to form (a) number of…**
It has been done (L. 11).

**Line 17: it isn't clear what you are "following up" on, you should either delete this, or provide further details.**
It has been done. "Follow up" has been replaced by "**this**" (L. 15).

**Line 18: monoterpene (singular).**
It has been done (L. 16).

**Line 20: since there is some previous data, you cannot state that there is a lack of experimental data, since lack denotes an absence.**
It has been done. "to fill the lack" has been replaced by "**to complete the few experimental data existing in the literature**"(L. 18).

**Line 25: are also, not also are. Precursors, not precursor.**
It has been done (L. 23).

**Line 35: Globally, there have been some changes since 1995. Is there a more up-to-date account of VOC emissions from the various sources?**
To our knowledge, no newer article gives an estimation of this ratio 90/10. This article is still cited in very recent papers such as Sindelarova et al., 2022 or Wu et al., 2020. Nevertheless, to provide a more up-to-date estimation of BVOC emission, we used a more recent paper by the same author (Guenther et al., 2012, L. 51).

**Line 53: For me, "proved" is a rather strong word to use, especially without a citation to back up your assertion. Scientifically, it is generally easier to disprove something, rather than prove it. Please consider a rephrase.**

It has been done. The sentence "β-caryophyllene has been proved to be the most emitted sesquiterpene" has been changed to "**β-caryophyllene is considered to be the most emitted sesquiterpene**" (L. 51).

**Line 81: requested is not the right word. Required?**

It has been done. We used the word "**required**" as the referee proposed (L. 81).

**Line 96: detail, not details.**

It has been done (L. 96).

**Line 120 (whole paragraph): Where are the data for these integrated band intensities?**

The references for $NO_2$, $HNO_3$ and $N_2O_5$ integrated band intensities (Rothman et al., 2003; Hjorth et al., 1987; Gordon et al., 2017, respectively) have been added (L.122-124). For terpinolene and β-caryophyllene, the IBIs have been measured experimentally in the frame of this study, because no data were available in the literature. The sentence "**(measured experimentally for this study)**" has been added (L. 122).

**Line 127: Reaction, not Reactor.**

It has been done (L. 128).

**Line 130: It is potentially confusing to talk about the reactor in the instrument in this sense, since you are dealing with other types of chamber reactors in this study. Would it be better to give it a more specific name such as a drift tube or an ion-molecule region?**

It has been done. "Reactor" has been changed to "**drift tube**" (L. 131).

**Line 136 (whole paragraph): Again, it is nice to be able to point to the actual data for these band intensities that you are referring to.**

The purpose of this paragraph is to explain how these values were obtained. No value for ON integrated band intensities was available in the literature when the experiments were conducted. To measure it, the IBI of two organic nitrate standards were measured (nitrooxypropanol and tert-butyl nitrate). The value we used in this study is the average of these two values. Therefore, this value is not coming from the literature but from experiments in our group. We think that furnishing the calibration plots is not relevant and would make the text more cumbersome.

**Line 144: It is not entirely clear, but it seems like what you are describing is not a rate exactly, but the extent to which the chamber contents have been diluted. Rephrase.**

It has been done. The sentence has been changed to "**the gas mixture in the chamber is diluted of max. 20%**" (L. 146).

**Line 146: Please describe how the dilution rate was measured.**

Because instrument sampling causes a pressure decrease in the chamber, pure air is continuously injected to maintain the pressure constant. The dilution rate was calculated by measuring the flow of pure air introduced into the chamber.

To make it clearer, the paragraph has been modified to: "**Because instruments sampling causes a pressure decrease in the chamber, pure air is continuously injected to maintain the pressure constant. The consequence is that the mixture is subject to a progressive dilution. The dilution rate was calculated thanks to the measurement of the pure air injection flow. For a typical flow rate of 1.7 L.min$^{-1}$, the gas mixture in the chamber is diluted of max. 20% after 3 hours of experiment. All data presented here were corrected from dilution.**"(L. 143-147).

**Line 172: were particle filters also employed? If not, what is the rationale for avoiding them?**
The IBBCEAS used in this chamber is an *in situ* instrument. It does not sample the mixture, but analyze it inside the chamber. The mirrors are directly in contact with the mixture. For more information about the configuration, it is precisely described in Fouqueau et al., 2020. It is thus impossible to use particle filters in this context.

Particles affect in two ways the measurement: by light absorption/scattering and by soiling the mirrors. First, in order to avoid the second first phenomenon, the mixing ratio of precursors were reduced as much as it is possible, to avoid a too high production of particles. In these conditions, no absorption or scattering was observed during the experiments. To avoid the soiling of the mirrors, they are flushed by nitrogen. The flush flow was determined to protect efficiently the mirrors and to avoid too much local dilution. This small local dilution was taken into account in the calculation of the reflectivity (Fouqueau et al., 2020).The sentence: "**In addition, the mirrors were flushed with Nitrogen to protect them from particle deposition.**" has been added (L. 175-176).

**Line 181: how small?**
"Small time" is a way to express the small step factor h, which is the base of Euler method to resolve differential equations. There is actually no rules to choose h value in this method. However in this case, we made the approximation that the value of 10s is small enough in comparison to the duration of the reaction (max. 1-2 minutes), and will allow a precise estimation.
The sentence "**, such as the time resolution used in this study,**" (L. 183) has been added in order to make this statement clearer.

**Lines 199 – 202: There are other recommendations besides IUPAC. There are other recommendations for this reaction: the recent recommendation of McGillen et al. (2020) 5.7E-11, who chose to accept the earlier recommendation of Calvert et al. 2015. This same reference was employed recently by Newland et al. 2021, which showed good consistency with other reference compounds used in this work suggesting that the uncertainty is really not so high.**
These recommendations were known but the uncertainty to the recommended value is very high and we considered it is overestimated. That's why we did not initially considered this value and proposed our own one. But following your comment, we have changed the rate constant and replaced it by the value recommended by Calvert and McGillen et al; 2020 (we changed 5.5 by 5.7). However, we reevaluated the uncertainty and calculated it as the mean value of the determinations available in the literature (Berndt et al., 1998; Benter et al., 1992; Lancar et al., 1991; Rahman et al., 1988; Atkinson et al., 1988, 1984a, b). The obtained value is: $k_{2,3\text{-dimethyl-2-butene}}$ = $(5.7 \pm 1.7) \times 10^{-11}$ cm$^3$ molecule$^{-1}$ s$^{-1}$. The same value and corresponding uncertainty was used by Newland et al., 2022.

The manuscript has been changed accordingly (L200-209): "**In this work, 2,3-dimethyl-2-butene was used as reference compound because of its well-known rate constant with NO$_3$ radicals. In absence of recommendation by IUPAC, the value recommended by Calvert et al., 2015 and by McGillen et al., 2020 was used. However, the uncertainty proposed by these recommendations is very high (150%) despite the fact that experimental determinations are in good agreement. Therefore, uncertainty was reevaluated and calculated as the mean value of the determinations available in the literature (Berndt et al., 1998; Benter et al., 1992; Lancar et al., 1991; Rahman et al., 1988; Atkinson et al., 1988, 1984a, b). The obtained value is: $k_{2,3\text{-dimethyl-2-butene}}$ = $(5.7 \pm 1.7) \times 10^{-11}$ cm$^3$ molecule$^{-1}$ s$^{-1}$. The same value and corresponding uncertainty was used by Newland et al., 2022. Finally, the uncertainty on kBVOC was calculated by considering the relative uncertainty corresponding to the statistical error on the linear regression (2σ) and the error on the reference rate constant.**"
Also, BVOC rate constants have been changed in the abstract (L. 19, mean of both absolute and relative determinations) and in Table 2 (relative determinations) to **$(6.0 \pm 3.8) \times 10^{-11}$ and $(1.8 \pm 1.4) \times 10^{-11}$ cm$^3$ molecule$^{-1}$ s$^{-1}$**, using the new reference compound rate constant. Lifetime presented in Table 6 have been recalculated with updated values.

**Line 207: (A) mechanistic study, or Mechanistic stud(ies) were conducted…**
It has been done (L. 211).

**Line 208: Past tense of leave: left**
It has been done (L. 213).

**Line 209: dark (for) (approximately)**
It has been done (L. 213).

**Line 209: it isn't purely wall-losses that you should be concerned about. What about other dark losses?**
The referee is right saying this. It was an omission, the dark period is achieved to verify both wall losses and dark losses. It has been modified in the text (L. 213): "**wall/dark losses**".

**Line 220: Product formation**
It has been change (L. 224).

**Line 234: Do you have any idea how accurate SIMPOL is for the classes of compound that you are applying it to?**
For terpinolene, most of the compounds have a Pvap>$10^{-6}$ atm, which is associated according to Pankow and Asher, 2008, to an error of max. 50% at 293.15 K. The heavier product has a Pvap of $10^{-7.9}$ atm, which is associated to an error of around 60%.
For β-caryophyllene, the vapor pressures of the products is a lot higher (between $10^{-7}$ and $10^{-10}$ atm) which is associated to an error between 50% and 80% respectively.
The sentence "**The uncertainty can reach 80% for $P^{vap}$ = 10$^{-10}$ atm.**" has been added (L. 257-258).

**Line 235: March (not Mars).**
It has been done (L. 247).

**Table 1: I don't like the formatting of BVOC name. Perhaps you can rotate the names by 90 degrees, which will allow you to avoid these inconsistent abbreviations.**
It has been done (Table 1).

**Line 259: Since you're just doing two different techniques, you might as well specify them in full.**
We are not sure to properly understand this remark, but if it is about the writing of the techniques in the table, the place is limited and the cases are smaller than the ones of the BVOC names. The cases have been merged to make the table clearer (Table 1).

**Figure 2 caption: there are no round marks, only square marks and triangles.**
It has been corrected (Figure 2 caption).

**Figure 3: It is apparent that the absolute measurements have a large amount of experimental variability. It would be instructive for the authors to suggests reasons why this may be the case. Is it possible that reaction times are sufficiently rapid that they are affected by mixing times and instrument response time?**
As mentioned L. 279, the reason to this is that we use low integration time (10s) for both IBB-CEAS and PTR-ToF-MS as the reaction is very fast. Nevertheless, the good agreement with previous studies and the fact that the dispersion is random and not systematic, suggest that the determination is accurate (the dispersion of experimental points has of course a consequence on the uncertainty). In addition,

such dispersion in experimental points was also observed for much slower reactions for which there was no issue about mixing time.

**Line 298 (whole paragraph): I find this comparative discussion of reactivity to be more confusing than it is educational. You mention the enhanced stability of the exocyclic double bond as evidence for the increased reactivity of terpinolene (which seems to be strange and unexpected). You also mention that the conjugated system of alpha terpinene leads to a stabler alkyl radical. If I am understanding this correctly (and there is a good chance that I am not), you are arguing that if the reactant or product is stabler, then k is bigger. So, I must insist that you state this argument more thoroughly. Are you trying to justify this on thermodynamic grounds? Something else? Let the readers know.**

The purpose of this paragraph is to explain the differences in the reactivity of the four molecules. It is obvious that the rate of the reaction depends on the activation energy which is linked to the stability of the transition state, not to the product one (here the alkyl radical). However, for electrophilic additions, it is often considered that as the organization of the atoms is very close in the transition state and in the product so that the Gibbs energy and the activation energy evolve in the same way when comparing similar reactions (Hammond postulate). Here, we considered that both the transition state and the product are stabilized by resonance. But to be more rigorous and to improve the clarity of the discussion following the referee's comment, the text has been changed as follows: **"α-terpinene appears to be much more reactive than γ-terpinene, due to the conjugation of double bonds which leads to a stabilization of the transition state by resonance. Here, terpinolene is almost twice more reactive than γ-terpinene, and this can be explained by the substitution of the exocyclic double bond which stabilizes the adduct" (L. 313-316).**

**Table 3: You acknowledge that there are potentially large uncertainties in these estimates. Do you expect that there errors that you are providing here would be symmetrical in each case. I think this would be surprising. Please can you explain how these errors were calculated?**

We agree that in theory, errors should not be symmetric but be provided in log scale (according to a log normal distribution of the population) to prevent from negative values. However, for practical purpose, in order to facilitate the use of the data, symmetric errors are given in most (if not all) kinetic studies.

Uncertainties on formation yields were calculated as the sum of the relative uncertainties on the product and the BVOC cross sections, and twice the standard deviation on the linear regression. The following sentence has been added (L. 227-229): **"Uncertainties on formation yields were calculated as the sum of the relative uncertainties on the product and the BVOC cross sections, and twice the standard deviation on the linear regression."**

Uncertainties on SOA yields were calculated as the sum of the relative errors on VOC concentrations measured by FTIR and the SOA concentration measured by SMPS. This sentence has been added in the manuscript (L.236).

Total ON yields (gas and particle phases) errors are the addition of the two relative errors of $Y_{ONp}$ and $Y_{ONg}$. A sentence has been added (L. 229-231): **"Organic nitrates have been measured both in gas and particle phase. Consequently, a total organic nitrate yield has been calculated, being the addition of these two yields. Their uncertainties were calculated as the sum of the relative uncertainties on gas and particle phase yields."**

Finally, $Y_{ONp, mass}$ / $Y_{SOA,mass}$ errors are the sum of relative errors on $Y_{ONp}$ and $Y_{SOA}$. A sentence has been added (L. 236-237): **"Knowing both the organic nitrate yield in particle phase and the total SOA yield, the ratio YONp, mass / YSOA,mass  has been calculated. Uncertainties were calculated as are the sum of relative errors on YONp and YSOA."**

**Table 3: Similar to Table 1: I don't like the formatting of BVOC name. Perhaps you can rotate the names by 90 degrees, which will allow you to avoid these inconsistent abbreviations.**

It has been done (Table 3).

**Lines 406 – 409: I agree with the authors about the sensitivity of SOA yields to various experimental parameters. Is it possible that [NO2] and radical concentration in general could also play an important role in SOA formation? It is important to note for example that although you have made specific efforts to reduce N2O5, you will inevitably have higher [NO2] than in a normal environment. Can the authors suggest a route towards understanding this SOA yield of these terpenoid compounds in a more fundamental way?**

We agree that $NO_2$ concentrations are significantly higher in our experiments than those usually encountered in the atmosphere. The main consequence of that is the potential formation of peroxynitrates from $RO_2+NO_2$ reactions. Some peroxynitrates were detected by PTR-MS but not by FTIR suggesting that their formation is a minor pathway (below 5% considering a detection limit of 15 ppb). Peroxynitrates were also not detected in the aerosol phase from FTIR analyses of filters

A precision has been added L.513-515: "**It should also be noticed that peroxynitrates (RO$_2$NO$_2$), which have a characteristic absorption in the IR region, were not detected in our experiments, neither in the gaseous phase, nor in the aerosol one. This suggests that that RO$_2$ + NO$_2$ reactions are minor pathways**." and L. 579-580: "**As for terpinolene, these compounds have not been detected in gas or particle phase, suggesting that the pathway is minor.**"

**Line 413 - 414: Although I just-about understand what you mean to say by this statement, I don't think this is a good way of saying it. Whether the slope was zero for a secondary product would very much depend on delta[alkene] over which the slope was calculated. Please consider a more robust rephrase of this statement (or delete it).**

The idea of this explanation is that, if the plot show a linear tendency, it can be explain by the fact that ON are primary products, and by the fact that potential ON secondary products are formed by the reaction of primary ON. The differentiation between primary and secondary ON is not possible with FTIR.

To be clearer in the explanation, the following sentence was added L. 416-417: "**as FTIR measurement cannot differentiate primary and secondary organic nitrates.**"

**Figure 6 caption: beta-caryophyllene is written incorrectly.**
It has been done (Figure 6).

**Lines 434 - 437: Since the yields of ONs are dependent upon the starting concentration of VOC, and since the concentration of VOC is higher than typical ambient concentrations, is it possible to obtain a good quantification of ON under ambient conditions?**

First, it is important to precise that the total ON yields (in both gas and aerosol phases) were observed not to be dependent upon the starting concentration of VOC. ON yield measured in particle phase is indirectly dependent on reactant concentration as it is dependent upon the amount on SOA generated, which is itself dependent on the amount of reactant. To sum up, the more reactant there is, the more SOA will be formed (due to a higher presence condensation nuclei) of and the more ON will partition towards the aerosol phase. We agree that because the yields of ONs in particle phase were not measured at the beginning of the oxidation (when aerosol content is representative of ambient air) of the but at the end of the experiment, they may be overestimated. A sentence has been added in the text: "**They are thus probably overestimated in comparison with real atmospheric conditions.**" (L. 436-437).

The proportion of ON in SOA calculated with these value can be compared to those obtained in field studies. ON are a major component of SOA (between 30 and 66 % for terpinolene and 80 % for β-caryophyllene). It is in good agreement with field studies measurement that showed that ON are major component of SOA (values are often higher than 50 %, Ng et al., 2017). Even if these organic nitrates

are not only products of NO3 chemistry, it is interesting to see that similar composition are found. This implies that the quantification of ON in our study should not be so far of the values under ambient conditions. This discussion is made from L. 448 to L. 458.

**Line 451: field studies**
It has been done (L. 453).

**Line 452: component(s)**
It has been corrected (L. 454).

**Line 454: In a region that was impacted by NO3 radical at night, one would expect elevated NOx concentrations. This would enhance nitrate yields in chemistries that were initiated by other oxidants such as OH, so I am not sure that this comparison with the field is very insightful.**
As said in this discussion (L. 448-458), in these conditions, ON can be formed by other reactions with OH radical or $O_3$. Nevertheless, this comparison support the major role of organic nitrates in SOA formation.

**Line 456: I don't think it really confirms anything. At best, I would suppose that it suggests that NO3 chemistry may contribute towards SOA production.**
The referee is right. The sentence "both the importance of $NO_3$ chemistry in SOA formation and" has been deleted and the sentence is now "**This result thus confirms the major contribution of organic nitrates in SOA formation.**" (L. 457-458).

**Line 460 - 461: Is this really true? Do you know that all classes of compounds are detected in an accurate and sufficiently sensitive way using this approach? Please justify this statement.**
The referee is right saying term accurate we too strong. It has been modified by "**double**" (L. 461).

**Table 4 caption: please provide a definition of the various intensities that you are listing in this table.**
The following sentence has been added L. 465: "**In Table 4, intensities were shown following this logic: The one or two most intense peak are noted "+++" and are usually at least one order of magnitude higher than the other ones. Peaks that are more intense than 10 counts are marked "++" and the other ones are noted "+".**"

**Sections 4.3.1. and 4.3.2.: this section is quite difficult to keep track of. I don't have any very constructive things to say here, only that it is difficult to maintain focus on this part of your paper.**
We are aware that these parts are very demanding and difficult part, and are consequently difficult to follow. We tried our best to be as concise as possible. We think that the description of the mechanism will lead to this kind of difficultness whatever the way it is presented, and if we are not removing information.

**Line 586: gas not gaz**
It has been done (L.603).

**References**

Calvert, J. G., Orlando, J. J., Stockwell, W. R., and Wallington, T. J.: The Mechanisms of Reactions Influencing Atmospheric Ozone, Oxford University Press, New York, 2015.

Fouqueau, A., Cirtog, M., Cazaunau, M., Pangui, E., Zapf, P., Siour, G., Landsheere, X., Méjean, G., Romanini, D., and Picquet-Varrault, B.: Implementation of an IBBCEAS technique in an atmospheric simulation chamber for in situ NO3 monitoring: characterization and validation for kinetic studies, 2020.

Fry, J. L., Draper, D. C., Barsanti, K. C., Smith, J. N., Ortega, J., Winkler, P. M., Lawler, M. J., Brown, S. S., Edwards, P. M., Cohen, R. C., and Lee, L.: Secondary Organic Aerosol Formation and Organic Nitrate Yield from NO3 Oxidation of Biogenic Hydrocarbons, Environ. Sci. Technol., 48, 11944–11953, https://doi.org/10.1021/es502204x, 2014.

Gordon, I. E., Rothman, L. S., Hill, C., Kochanov, R. V., Tan, Y., Bernath, P. F., Birk, M., Boudon, V., Campargue, A., Chance, K. V., Drouin, B. J., Flaud, J.-M., Gamache, R. R., Hodges, J. T., Jacquemart, D., Perevalov, V. I., Perrin, A., Shine, K. P., Smith, M.-A. H., Tennyson, J., Toon, G. C., Tran, H., Tyuterev, V. G., Barbe, A., Császár, A. G., Devi, V. M., Furtenbacher, T., Harrison, J. J., Hartmann, J.-M., Jolly, A., Johnson, T. J., Karman, T., Kleiner, I., Kyuberis, A. A., Loos, J., Lyulin, O. M., Massie, S. T., Mikhailenko, S. N., Moazzen-Ahmadi, N., Müller, H. S. P., Naumenko, O. V., Nikitin, A. V., Polyansky, O. L., Rey, M., Rotger, M., Sharpe, S. W., Sung, K., Starikova, E., Tashkun, S. A., Auwera, J. V., Wagner, G., Wilzewski, J., Wcisło, P., Yu, S., and Zak, E. J.: The HITRAN2016 molecular spectroscopic database, Journal of Quantitative Spectroscopy and Radiative Transfer, 203, 3–69, https://doi.org/10.1016/j.jqsrt.2017.06.038, 2017.

Guenther, A. B., Jiang, X., Heald, C. L., Sakulyanontvittaya, T., Duhl, T. R., Emmons, L. K., and Wang, X.: The Model of Emissions of Gases and Aerosols from Nature version 2.1 (MEGAN2.1): an extended and updated framework for modeling biogenic emissions, Geosci. Model Dev., 1471–1492, 2012.

Hjorth, J., Ottobrini, G., Cappellani, F., and Restelli, G.: A Fourier transform infrared study of the rate constant of the homogeneous gas-phase reaction nitrogen oxide (N2O5) + water and determination of absolute infrared band intensities of N2O5 and nitric acid, J. Phys. Chem., 91, 1565–1568, https://doi.org/10.1021/j100290a055, 1987.

McGillen, M. R., Carter, W. P. L., Mellouki, A., Orlando, J. J., Picquet-Varrault, B., and Wallington, T. J.: Database for the kinetics of the gas-phase atmospheric reactions of organic compounds, 12, 1203–1216, https://doi.org/10.5194/essd-12-1203-2020, 2020.

Newland, M. J., Ren, Y., McGillen, M. R., Michelat, L., Daële, V., and Mellouki, A.: NO3 chemistry of wildfire emissions: a kinetic study of the gas-phase reactions of furans with the NO3 radical, 22, 1761–1772, https://doi.org/10.5194/acp-22-1761-2022, 2022.

Ng, N. L., Brown, S. S., Archibald, A. T., Atlas, E., Cohen, R. C., Crowley, J. N., Day, D. A., Donahue, N. M., Fry, J. L., Fuchs, H., Griffin, R. J., Guzman, M. I., Herrmann, H., Hodzic, A., Iinuma, Y., Jimenez, J. L., Kiendler-Scharr, A., Lee, B. H., Luecken, D. J., Mao, J., McLaren, R., Mutzel, A., Osthoff, H. D., Ouyang, B., Picquet-Varrault, B., Platt, U., Pye, H. O. T., Rudich, Y., Schwantes, R. H., Shiraiwa, M., Stutz, J., Thornton, J. A., Tilgner, A., Williams, B. J., and Zaveri, R. A.: Nitrate radicals and biogenic volatile organic compounds: oxidation, mechanisms, and organic aerosol, Atmos. Chem. Phys., 2103–2162, 2017.

Pankow, J. F. and Asher, W. E.: SIMPOL.1: a simple group contribution method for predicting vapor pressures and enthalpies of vaporization of multifunctional organic compounds, Atmos. Chem. Phys., 2773–2796, 2008.

Rothman, L. S., Barbe, A., Chris Benner, D., Brown, L. R., Camy-Peyret, C., Carleer, M. R., Chance, K., Clerbaux, C., Dana, V., Devi, V. M., Fayt, A., Flaud, J.-M., Gamache, R. R., Goldman, A., Jacquemart, D., Jucks, K. W., Lafferty, W. J., Mandin, J.-Y., Massie, S. T., Nemtchinov, V., Newnham, D. A., Perrin, A., Rinsland, C. P., Schroeder, J., Smith, K. M., Smith, M. A. H., Tang, K., Toth, R. A., Vander Auwera, J., Varanasi, P., and Yoshino, K.: The HITRAN molecular spectroscopic database: edition of 2000 including updates through 2001, Journal of Quantitative Spectroscopy and Radiative Transfer, 82, 5–44, https://doi.org/10.1016/S0022-4073(03)00146-8, 2003.

Sindelarova, K., Markova, J., Simpson, D., Huszar, P., Karlicky, J., Darras, S., and Granier, C.: High-resolution biogenic global emission inventory for the time period 2000–2019 for air quality modelling, 14, 251–270, https://doi.org/10.5194/essd-14-251-2022, 2022.

Wu, K., Yang, X., Chen, D., Gu, S., Lu, Y., Jiang, Q., Wang, K., Ou, Y., Qian, Y., Shao, P., and Lu, S.: Estimation of biogenic VOC emissions and their corresponding impact on ozone and secondary organic aerosol formation in China, 2020.

**Answer to Anonymous Referee #2**

First of all, the authors would like to thank the anonymous referee for this discussion and its constructive comments, corrections and suggestions that ensued. We have carefully replied to all its comments and the paper has been improved following its recommendations. Answers have also been provided for all comments and changes have been performed accordingly. Please find below the answers to the comments:

**General Comments**

**This manuscript describes an experimental study of the reactions of a monoterpene and sesquiterpene with NO3 radicals. Rate constants were measured in a glass chamber using absolute and relative rate methods. Product and mechanism studies were conducted in a stainless-steel chamber and gas-phase products were analyzed online using a proton transfer reaction-mass spectrometer with a H2O+ and NO+ ion source. Gas and aerosol products were analyzed by infrared spectroscopy to quantify nitrate compounds. Particle size and volume concentrations were monitored with a scanning mobility particle sizer. The study is relevant to understanding the nighttime formation of organic nitrates from VOC oxidation, which can impact SOA and ozone formation. The measured rate constants generally agree with those measured previously, thereby giving confidence to the reported values. The study also provides new yields of acetone, organic nitrates, and SOA; and the reaction mechanisms proposed for each compound seem to explain the observed products quite well. Overall, I think the measurements were well done, the interpretation of the date is accurate, and the paper is clearly and concisely written. I think the manuscript is publishable in ACP, and I have only a few minor comments.**

**Specific Comments**

**Line 95: Please define IBBCEAS.**
The acronym IBBCEAS has been previously define in the introduction of "2 Experimental section" (line 82).

**Line 215: Can't you use ion-molecule reaction rate theory to estimate rate constants for ionization and then quantify products?**
The application of ion-molecule reaction rate theory for estimating rate constants for ionization in the frame of a PTR-MS has been mainly developed for $H_3O^+$ ionization (and in a minor extent for $O_2^+$ and $NH_4^+$, Bhatia et al., 2020; Strekowski et al., 2019). To our knowledge, it has never been used in a $NO^+$ oxidation study.
The study of Sekimoto and Koss, 2021 shows that calculated ion-molecule rate constants for $H_3O^+$ + VOC may have an accuracy of 10 % in comparison to the measured ones. These constants are used in a frame of PTR-MS VOC sensitivity calculation, which can be estimated with an accuracy of 20-50 % without direct standard calibration, depending on the physical conditions in the drift tube. These sensitivities can be significantly worst for some compounds, and particularly if their fragmentation is unknown. In the case of our study, the composition of the analyzed mixtures is complex and the fragmentation behavior of the products is not completely understood due to the high quantity of peaks and the similar structure of the products (which have similar fragments). In this context, the quantification would have been associated to a high uncertainty.
Also, the use $NO^+$ as ionization agent, which is still not investigated with this technique, lead to different ionization pathways (as explained L. 132-136), and consequently to very complex rate constant estimations. The application of this theory is thus a major challenge in this technique.
Considering these limitations, we preferred to not use this method to quantify the products, and mainly use the PTR-MS as a qualitative technics.

**Line 220: What products do you mean? Acetone? On line 215 you state that you can't quantify products because of the lack of standards.**
In this sentence, "products" was referring to the products that are measured by FTIR, which allow their quantification, i.e. total organic nitrates and acetone. The sentence has been modified to: "**When products can be quantified by FTIR concentration is measurable by FTIR, product their formation yields were calculated by plotting their molecular concentration of product against the reacted BVOC molecular concentration and by calculating the slope at the origin.**" (L.224-225).

**Line 355: Stating that the SOA yields are below 90% is not very useful, since that means they could be anywhere from 0 to 89%. I suggest giving the actual range of yields.**
The sentence has been changed to "**between 50% and 90%**" (L. 357).

**Line 376: Since you know the VOC-NO3 rate constant and approximate NO3 radical concentration, it seems that you can calculate the reaction lifetime and compare that to the mixing timescale. That would be useful support for the explanation given here.**

We considered doing such a calculation, however $NO_3$ was not monitored during mechanistic experiments and $N_2O_5$ concentration was below the detection limit preventing from estimating $NO_3$ concentration. Therefore, the calculation of reaction lifetime is impossible.

**Figure 8. I think NO3 addition occurs preferentially to form the tertiary alkyl radical, so wouldn't it be better to show that reaction pathway?**

We agree that the formation of the tertiary radical is expected to be the major pathway. However, because the formation of hydroxy-nitrate was observed and can only be explained by the formation of the secondary radical, we decided to show this pathway, not as the major one but just to allow explaining the hydroxy-nitrate formation.
In conclusion, if the tertiary alkyl radical coming from the addition on the endocyclic radical is shown, the figure will not contain the formation of hydroxy-nitrate compound, which is a key product in this study.

**Neither Figure 8 nor Figure 9 consider possible isomerization of alkoxy radicals, and assume instead only reaction with O2 or decomposition. This assumption might be supported using SAR calculations of Vereecken & Peeters, PCCP, 2009 & 2010, although results from Aschmann et al. JPCA 2011 for cycloalkoxy radicals indicate that for bcaryophyllene some isomerization should occur.**

This was an omission but we agree that we have to discuss this point. The reason of this omission was that no compounds coming from isomerization were detected, and it was considered minor by Vereecken & Peeters, 2009 & 2010 SAR. On Figure 8 and 9, only the products that were detected are shown. Products coming from this pathway were searched but none was found. Nevertheless, this reaction lead to the formation of heavy functionalized products that can be difficult to measure with PTR-MS for two reasons: (i) it cannot measure too heavy products, which is probably the case for isomerization products of β-caryophyllene, and (ii) these compounds can be probably found largely in particle phase. No analysis at the molecular scale was conducted in the particle phase during our experiments. Indeed, in this study we only measure the total organic nitrates in the aerosol phase from their IR absorption band. Nitrates formed by this pathway will thus not be differenced. The occurrence of this pathway is thus not in disagreement with the observation of high SOA formation.

The following text has been added L. 655-664: "**Products coming from isomerization were not detected in this study. Even though it is considered as minor pathway by Vereecken and Peters, 2009 calculation, it was proved to be possible in Aschmann et al., 2012 for cycloalkoxy radicals.**

Isomerization could thus occur for β-caryophyllene. Products coming from this pathway were searched but none was found. Nevertheless, this reaction lead to the formation of heavy functionalized products that can be difficult to measure with PTR-MS for two reasons: (i) it cannot measure too heavy products, which is probably the case for isomerization products of β-caryophyllene, and (ii) these compounds can be found largely in particle phase. No analysis at the molecular scale was conducted in the particle phase during our experiments. Indeed, in this study we only measure the total organic nitrates in the aerosol phase from their IR absorption band. Nitrates formed by this pathway will not be differenced with other ones. The occurrence of this pathways is thus not in disagreement with the observation of high SOA formation."

In addition, the following citation has been added:

"**Aschmann, S. M., Arey, J., and Atkinson, R.: Kinetics and Products of the Reactions of OH Radicals with Cyclohexene, 1-Methyl-1-cyclohexene, cis-Cyclooctene, and cis-Cyclodecene, https://doi.org/10.1021/jp307217m, 2012.**"

**Line 588: Similar to Comment 3, it is not very useful to state that SOA yields are <100%.**
It has been corrected to "**between 50 and 90 % and not 100 %**" (L. 606-607).

**References**

Bhatia, M., Biasioli, F., Cappellin, L., Piseri, P., and Manini, N.: Ab initio calculation of the proton transfer reaction rate coefficients to volatile organic compounds related to cork taint in wine, 55, e4592, https://doi.org/10.1002/jms.4592, 2020.

Sekimoto, K. and Koss, A.: Modern mass spectrometry in atmospheric sciences: Measurement of volatile organic compounds in the troposphere using proton-transfer-reaction mass spectrometry, 56, https://doi.org/10.1002/jms.4619, 2021.

Strekowski, R. S., Alvarez, C., Petrov-Stojanović, J., Hagebaum-Reignier, D., and Wortham, H.: Theoretical chemical ionization rate constants of the concurrent reactions of hydronium ions (H3 O+ ) and oxygen ions (O  2 +   ) with selected organic iodides, J Mass Spectrom, 54, 422–428, https://doi.org/10.1002/jms.4349, 2019.

---

## Author Response (AR2)

**Answers to Editor**

First of all, the authors would like to thank the editor for the suggestions. We have carefully replied to all its comments and the paper has been improved following its recommendations. Answers have also been provided for all comments and changes have been performed accordingly. Please find below the answers to the comments:

**Line 175 : "nitrogen"**

It has been corrected (L.176)

**Figure 5 : It would be nice to have the literature data (Fry et al., 2014; Wu et al., 2021) for comparison in the plots.**

The figure has been modified to show SOA yields taken from Wu et al., 2021; Jaoui et al., 2013 and Fry et al., 2014. The following sentence has been added in the caption of Fig. 5: "**For β-caryophyllene literature values are presented by squared marks.**"

**Comment : Please discuss the findings by Wu et al., ACP, 2021 in comparison with your results.**

The author want to thank the editor for this suggestion. The following discussions have been added :

"**Finally, Wu et al., 2021 studied the impact of photolysis on NO$_3$-generated SOA for β-caryophyllene. They measured a final SOA yield (110%) and provided particle-phase composition analysis, showing a major impact of organic nitrates. No yield plot was provided and β-caryophyllene concentration could not have been monitored using quadrupole-PTR-MS, due its m/z ratio outside of mass transmission range.**" (L. 69-74)

"**Finally, Wu et al., 2021 studied the photolytically induced ageing of NO$_3$-initated SOA. To do so, they first have generated SOA by reacting β-caryophyllene and NO$_3$. Experiment was conducted with 50 ppb of precursor, and a final SOA yield of 110 % is calculated. Two issues are pointed out: first, they could not monitor β-caryophyllene with a quadrupole-PTR-MS, because it was out of the mass transmission range for quantitative measurement. The method used to calculate its concentration is then not explained, but it is probably associated with a larger uncertainty. Second, a concentration of more than 200 ppb of N$_2$O$_5$ is injected during approx. 10s. As explained before, it can lead to an overestimation of SOA yield, and thus explain the observed difference. Nevertheless, the mean diameter of size distribution measured in this study is between 229 and 266 nm, which is in good agreement with the one measured here (between 225 and 246 nm in the end of the oxidation). This study showed no evaporation of SOA during a dark ageing, which confirm the fact that SOA concentrations are stable here, after the oxidation. They showed a major impact on SOA composition of photolysis in the case of β-caryophyllene, but, no photolysis was conducted in our study.**" (L. 409-421)

"**The study of Wu et al., 2021 carried out an identification of SOA composition. Hundreds of molecular compositions were identified using both FIGAERO-CIMS and EESI-TOF techniques, including a large majority of organic nitrates. C15 monomers are major products, as shown also in our study. C30 dimers have also been detected, but they are heavy products and out of the range of PTR-ToF-MS used in our study. In addition, the amount of dimers detected in particle phase can be explained by the reaction of hydroxynitrates with carbonyl compounds, via an acid-catalyzed particle-phase reaction leading to the formation of acetal dimers and trimers, as shown in Claflin and Ziemann, 2018.
This study is in good agreement with the determination of organic nitrates in particle phase: a large amount of organic nitrates was detected, which confirm their prominence in SOA formation for β-caryophyllene + NO$_3$ system. Most of the products were too heavy to be detected in our study, but two major ones are C$_{15}$H$_{24}$O$_2$**

**(MW=236 g/mol) and $C_{15}H_{25}NO_5$ (MW=298 g/mol). They have been identified here as opening ring products. It confirms the importance of these two products in β-caryophyllene + $NO_3$ chemistry.**" (L. 634-645)

The following references have been added :

"**Claflin, M. S. and Ziemann, P. J.: Identification and Quantitation of Aerosol Products of the Reaction of β-Pinene with NO3 Radicals and Implications for Gas- and Particle-Phase Reaction Mechanisms, J. Phys. Chem. A, 122, 3640–3652, https://doi.org/10.1021/acs.jpca.8b00692, 2018.**"

"**Wu, C., Bell, D. M., Graham, E. L., Haslett, S., Riipinen, I., Baltensperger, U., Bertrand, A., Giannoukos, S., Schoonbaert, J., El Haddad, I., Prevot, A. S. H., Huang, W., and Mohr, C.: Photolytically induced changes in composition and volatility of biogenic secondary organic aerosol from nitrate radical oxidation during night-to-day transition, 21, 14907–14925, https://doi.org/10.5194/acp-21-14907-2021, 2021.**"

---

## Author Response (AR3)

**Answers to Editor #2**

First of all, the authors would like to thank the editor for the suggestions and changes have been performed accordingly. Please find below the answers to the comments:

**Comment: Please improve the english of the discussion of the Wu et al. paper to avoid any misunderstandings.**

Newest modifications are shown in red in the following text.

"**Finally, Wu et al., 2021 studied the impact of photolysis on $NO_3$-generated SOA for β-caryophyllene. They measured a final SOA yield (110%) and provided particle-phase composition analysis, showing a major impact of organic nitrates. Nevertheless, neither $Y_{SOA}$ vs. $M_0$ graph nor SOA model parameters were provided. In addition, β-caryophyllene concentrations could not be measured by the quadrupole-PTR-MS, due to its m/z ratio outside mass transmission range.**" (L. 69-74)

"**Finally, Wu et al., 2021 studied the photolytically induced ageing of $NO_3$-initated SOA. In order to fulfil this aim, they first have generated SOA by reacting β-caryophyllene and $NO_3$. One experiments was conducted with 50 ppb of precursor, and a final SOA yield of 110 % is calculated. Two issues are pointed out: firstly, they could not monitor β-caryophyllene with a quadrupole-PTR-MS, because its m/z ratio was out of the range for quantitative measurement. The method used to calculate its concentration is then not explained, but it is probably associated with a larger uncertainty. Secondly, a concentration of more than 200 ppb of $N_2O_5$ is injected during approx. 10s. As explained before, it can lead to a SOA yield overestimation, and thus explain the observed differences. Nevertheless, the mean diameter of size distribution measured in this study is between 229 and 266 nm, which is in good agreement with the ones measured here (between 225 and 246 nm in the end of the oxidation). This study showed no evaporation of SOA during a dark ageing, which agrees with the fact that SOA concentrations are stable here, after the oxidation.**" (L. 409-421). The last sentence has been deleted.

"**The study of Wu et al., 2021 carried out an identification of SOA composition. Particle-phase molecular composition was identified using both FIGAERO-CIMS and EESI-TOF techniques. A large majority of organic nitrates were detected. C15 monomers are major products, as shown also in our study. C30 dimers have also been detected, but they are heavy products and out of the PTR-ToF-MS mass-to-charge ratio range used in our study. In addition, the amount of dimers detected in particle phase can be explained by the reaction of hydroxynitrates with carbonyl compounds, via an acid-catalyzed particle-phase reaction leading to the formation of acetal dimers and trimers, as shown in Claflin and Ziemann, 2018.**
**This study is in good agreement with the determination of organic nitrates in particle phase: a large amount of organic nitrates was detected, which confirm their prominence in β-caryophyllene + $NO_3$ SOA formation. Most of the products were too heavy to be detected in our study, but two major ones are $C_{15}H_{24}O_2$ (MW=236 g/mol) and $C_{15}H_{25}NO_5$ (MW=298 g/mol). They have been identified here as opening ring products. It confirms the importance of these two products in β-caryophyllene + $NO_3$ chemistry.**" (L. 634-645)